# Beyond the MEP Pathway: A novel kinase required for prenol utilization by malaria parasites

Marcell Crispim[1,2☯], Ignasi Bofill Verdaguer[1☯], Agustín Hernández[3], Thales Kronenberger[4,5,6], Àngel Fenollar[2], Lydia Fumiko Yamaguchi[7], María Pía Alberione[2], Miriam Ramirez[2], Sandra Souza de Oliveira[7], Alejandro Miguel Katzin[1]*, Luis Izquierdo[2,8]*

**1** Department of Parasitology, Institute of Biomedical Sciences of the University of São Paulo, São Paulo, Brazil, **2** Barcelona Institute for Global Health (ISGlobal), Hospital Clínic-Universitat de Barcelona, Barcelona, Spain, **3** Center for Biological and Health Sciences, Integrated Unit for Research in Biodiversity (BIOTROP-CCBS), Federal University of São Carlos, São Carlos, Brazil, **4** Institute of Pharmacy, Pharmaceutical/Medicinal Chemistry and Tuebingen Center for Academic Drug Discovery, Eberhard Karls University Tübingen, Tübingen, Germany, **5** School of Pharmacy, Faculty of Health Sciences, University of Eastern Finland, Kuopio, Finland, **6** Excellence Cluster "Controlling Microbes to Fight Infections" (CMFI), Tübingen, Germany, **7** Institute for Technological Research of São Paulo State, Nutabes, São Paulo, Brazil, **8** Centro de Investigación Biomédica en Red de Enfermedades Infecciosas (CIBERINFEC), Barcelona, Spain

☯ These authors contributed equally to this work.
* amkatzin@icb.usp.br (AMK); luis.izquierdo@isglobal.org (LI)

**Data Availability Statement:** All relevant data are within the manuscript and its Supporting information.

## Abstract

A proposed treatment for malaria is a combination of fosmidomycin and clindamycin. Both compounds inhibit the methylerythritol 4-phosphate (MEP) pathway, the parasitic source of farnesyl and geranylgeranyl pyrophosphate (FPP and GGPP, respectively). Both FPP and GGPP are crucial for the biosynthesis of several essential metabolites such as ubiquinone and dolichol, as well as for protein prenylation. Dietary prenols, such as farnesol (FOH) and geranylgeraniol (GGOH), can rescue parasites from MEP inhibitors, suggesting the existence of a missing pathway for prenol salvage via phosphorylation. In this study, we identified a gene in the genome of *P. falciparum*, encoding a transmembrane prenol kinase (PolK) involved in the salvage of FOH and GGOH. The enzyme was expressed in *Saccharomyces cerevisiae*, and its FOH/GGOH kinase activities were experimentally validated. Furthermore, conditional knockout parasites (Δ-PolK) were created to investigate the biological importance of the FOH/GGOH salvage pathway. Δ-PolK parasites were viable but displayed increased susceptibility to fosmidomycin. Their sensitivity to MEP inhibitors could not be rescued by adding prenols. Additionally, Δ-PolK parasites lost their capability to utilize prenols for protein prenylation. Experiments using culture medium supplemented with whole/delipidated human plasma in transgenic parasites revealed that human plasma has components that can diminish the effectiveness of fosmidomycin. Mass spectrometry tests indicated that both bovine supplements used in culture and human plasma contain GGOH. These findings suggest that the FOH/GGOH salvage pathway might offer an alternate source of isoprenoids for malaria parasites when *de novo* biosynthesis is inhibited. This study also identifies a novel kind of enzyme related to isoprenoid metabolism.

**Funding:** MC and IBV are fellows from the Fundação de Amparo à Pesquisa do Estado de São Paulo (FAPESP); MC FAPESP process numbers: 2020/14897-6 and 2018/02924-9; IBV FAPESP process number: 2019/13419-6. This work was supported by FAPESP process number: 2017/22452-1 and 2014/10443-0, awarded to AMK and AHL respectively, Coordenação de Aperfeiçoamento de Pessoal de Nível Superior (CAPES) and Conselho Nacional de Desenvolvimento Científico e Tecnológico (CNPq). Institut de Salut Global (ISGlobal) is supported by the Spanish Ministry of Science and Innovation through the Centro de Excelencia Severo Ochoa 2019-2023 Program (grant number CEX2018-000806-S), and the Generalitat de Catalunya through the CERCA Program. This work is part of the ISGlobal Program on the Molecular Mechanisms of Malaria, partially supported by the Fundación Ramón Areces. LI receives support by PID2019-110810RB-I00 and PID2022-137031OB-I00 grants from the Spanish Ministry of Science & Innovation. MPA is supported by a FI Fellowship from the Generalitat de Catalunya supported by Secretaria d'Universitats i Recerca de la Generalitat de Catalunya and Fons Social Europeu (2021 FI_B 00470) and AF is supported by a Becas de Formación del Profesorado Universitario (FPU Fellowship) from the Spanish Ministry of Universities (FPU20-04484). TK is funded by the Exzellenzcluster „Kontrolle von Mikroorganismen zur Bekämpfung von Infektionen" (CIMF) and TüCAD2. CIMF and TüCAD2 are funded by the Federal Ministry of Education and Research (BMBF) and the Baden-Württemberg Ministry of Science as part of the Excellence Strategy of the German Federal and State Governments. The funders had no role in study design, data collection and analysis, decision to publish, or preparation of the manuscript.

**Competing interests:** The authors have declared that no competing interests exist.

## Author summary

Falciparum malaria is a potentially fatal disease caused by the parasite *Plasmodium falciparum*. Antimalarials such as fosmidomycin and clindamycin target a critical pathway in the parasite, crucial for producing certain substances essential for the parasite's survival, particularly phosphorylated isoprenoids. However, the limited effectiveness of these drugs in clinical trials for malaria treatment underscores the need for further related studies. Previous *in vitro* experiments have demonstrated that the parasite can utilize unphosphorylated isoprenoids, namely farnesol and geranylgeraniol, if they are present in the external environment. Thus, these substances act as antidotes, rendering the parasite resistant to both fosmidomycin and clindamycin. This study reveals for the first time that geranylgeraniol naturally occurs in the human body. Additionally, we have identified a novel enzyme, prenol kinase, which enables the parasite to use these unphosphorylated isoprenoids by converting them into their metabolically active phosphorylated counterparts. Parasites lacking the prenol kinase gene remain viable but become more susceptible to the effects of fosmidomycin, even in the presence of farnesol or geranylgeraniol. These findings suggest that the scavenging of unphosphorylated isoprenoids by the parasite might supplement its isoprenoid needs when the endogenous production is inhibited by drugs like fosmidomycin or clindamycin.

## 1. Introduction

*Plasmodium falciparum* causes the most severe form of human malaria, a parasitic disease with a high global burden. In 2021, the World Health Organization reported an estimated 247 million cases of malaria and 619,000 malaria-related deaths, with the majority occurring among children and pregnant women in Sub-Saharan Africa. In 2021, 96% of all malaria-related deaths occurred in this region. Resistance to current antimalarial drugs is a significant challenge for malaria control, leading to increased morbidity and mortality [1]. Therefore, the identification and development of novel antimalarial therapies are urgently needed.

The ancestor of apicomplexan parasites underwent endosymbiosis with an alga and thus, possess a non-photosynthetic plastid called the apicoplast [2] which contains the targets of some of the current antimalarial drugs in use [3,4]. The most extensively studied biological process in the apicoplast is isoprenoid biosynthesis via the methylerythritol phosphate (MEP) pathway (Fig 1). Unlike animals, which use the mevalonate (MVA) pathway, the MEP pathway condenses pyruvate and glyceraldehyde 3-phosphate to produce 5-carbon isoprene units, isopentenyl pyrophosphate (IPP) and dimethylallyl pyrophosphate (DMAPP). IPP and DMAPP are enzymatically condensed in geranyl pyrophosphate (10 carbon), farnesyl pyrophosphate (FPP, 15 carbon), and geranylgeranyl pyrophosphate (GGPP, 20 carbon) [5,6]. These metabolites are essential for protein farnesylation and geranylgeranylation (Fig 1). The parasite also produces longer polyprenyl pyrophosphates for the biosynthesis of ubiquinone-8,9 and dolichols, which are mitochondrial cofactors and lipid carriers for sugar transport in protein glycosylation, respectively [7–13]. Isoprenoids produced by the MEP pathway are thus involved in various essential parasitic processes, such as mitochondrial activity and post-translational modification of proteins. The antimalarial drug fosmidomycin inhibits the MEP pathway by targeting the enzyme 1-deoxy-D-xylulose 5-phosphate reductoisomerase (DXR), which

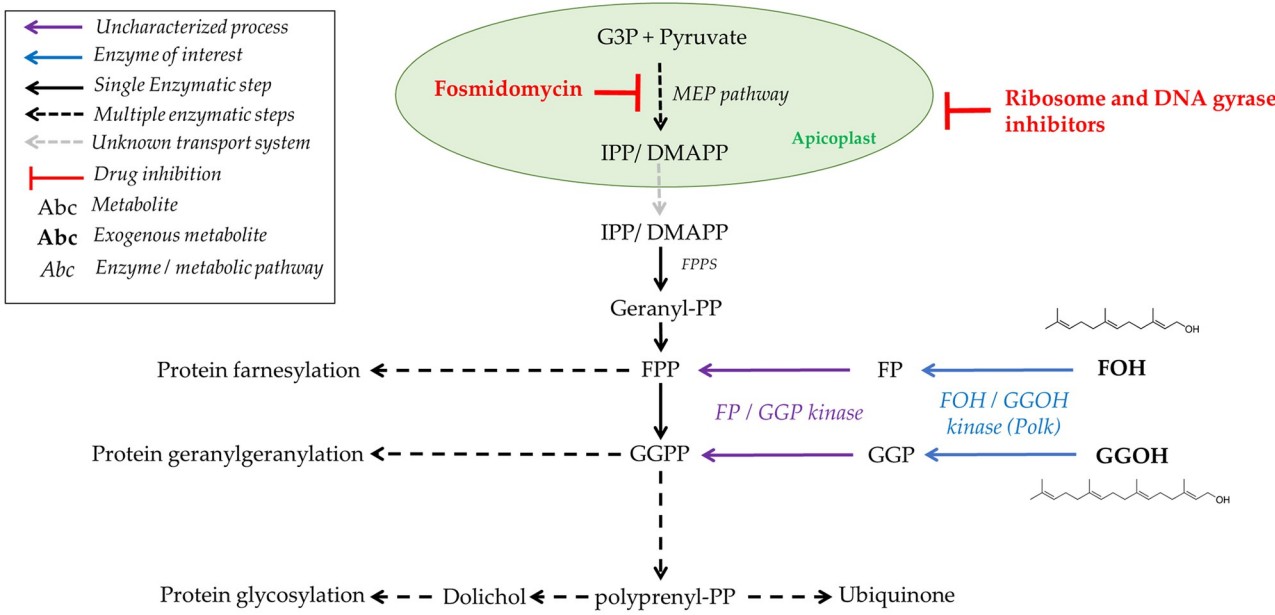

**Fig 1. Isoprenoid sources and distribution in malaria parasites.** The figure illustrates the sources and distribution of isoprenoids in malaria parasites. The figure includes the biosynthesis of isoprenoids via the MEP pathway, starting with the condensation of glyceraldehyde-3-phosphate (G3P) and pyruvate, and leads to the formation of isopentenyl pyrophosphate (IPP) and dimethylallyl pyrophosphate (DMAPP), and their subsequent condensation to form geranyl pyrophosphate (GPP), farnesyl pyrophosphate (FPP), and geranylgeranyl pyrophosphate (GGPP). These longer isoprenoids are essential for the biosynthesis of ubiquinone, dolichol, and for protein prenylation. The figure also shows the targets of fosmidomycin and ribosome inhibitors in the parasite. The chemical structures of FOH and GGOH are represented.

converts 1-deoxy-D-xylulose 5-phosphate (DXP) to MEP (Fig 1) [14]. Similarly, classic bacterial ribosome inhibitors (hereafter referred as ribosome inhibitors), such as azithromycin, doxycycline, or clindamycin, as well as DNA gyrase inhibitors [15] can indirectly inhibit isoprenoid biosynthesis by interfering with apicoplast biogenesis. Treated parasites transmit defective organelles to their progeny, leading to a delayed death effect in which the parasites exposed to these drugs cease growth approximately 48 hours after the onset of treatment [16–18]. Until recently, it was assumed that these drugs completely inhibited apicoplast formation and all the metabolic processes associated with this organelle. However, recent studies have found that treatment with ribosome inhibitors leads to the fragmentation of the apicoplast into vesicles. Whereas these vesicles do not maintain active the MEP pathway, they still contain other important enzymes [19].

Parasites can be grown indefinitely *in vitro* when exposed to fosmidomycin or ribosome inhibitors if there is an exogenous source of IPP. Additionally, parasites with impaired isoprenoid biosynthesis can be partially rescued by the addition of FPP/farnesol (FOH) and GGPP/geranylgeraniol (GGOH), but not isopentenol, geraniol, octaprenol, nonaprenol or dolichols [16–18,20]. Additional studies characterized the molecular and morphological phenotype of parasites exposed to ribosome inhibitors in order to investigate their isoprenoid requirements. Additionally, metabolomic profiling revealed that the lack of ubiquinone and dolichol biosynthesis is not the primary cause of death, but rather the disruption of the digestive vacuole function [18]. It is now understood that the loss of protein prenylation interferes with vesicular trafficking and ultimately affects *P. falciparum*'s feeding, leading to their death. In fact, the sort of prenylated proteins in malaria parasites includes Ras, Rho, and Rap small GTPases, which are involved in cellular signalling and intracellular trafficking [20,21].

Mechanistically, these processes rely on transferases that attach FPP or GGPP moieties to the C-terminal cysteine residues of proteins containing a conserved motif for prenylation, CAAX (C = cysteine, A = aliphatic amino acid, X = diverse terminal residue) [22].

Several clinical trials using fosmidomycin to treat malaria failed, mostly due to poor antimalarial efficacy [23]. Additionally, the combination of fosmidomycin plus clindamycin was also unsuccessful in clinics, with no clear evidence of mutations related to its resistance [24,25]. Thus, the failure of these therapies may be due to the intrinsic mechanisms of the parasite or the pharmacokinetics of fosmidomycin [26]. Therefore, it is crucial to conduct further research on isoprenoid metabolism in the parasite to develop effective antimalarial treatments targeting these pathways.

An unresolved matter regarding isoprenoid metabolism in *Plasmodium* is related to classic drug-rescue assays that employ the prenols FOH and GGOH [16,18,20,27–29]. There is no evidence that prenyl synthases/transferases would preferably bind prenols, or any other polyprenyl derivative, rather than polyprenyl-PP. Prenols may orient parallel to the membrane, while polyprenyl pyrophosphates favour a perpendicular orientation, making their pyrophosphate moieties physically available for interaction with polyprenyl transferases and synthases [30–32]. Therefore, a FOH/GGOH salvage pathway was proposed to exist, acting via phosphorylation to incorporate these molecules into the major isoprenoid metabolism [32]. Our group has previously characterized the transport of FOH and GGOH in *P. falciparum*, and observed that these prenols are phosphorylated, condensed into longer isoprenoids, and incorporated into proteins and dolichyl phosphates [20].

Prenol phosphorylation has been biochemically demonstrated to occur in membrane extracts of animal and plant tissues, as well as in archaea [32]. In 1998, Bentinger *et al*. reported the first evidence of this pathway in mammals, observing the conversion of FOH to farnesyl monophosphate (FP) in the 10,000 x *g* supernatant of rat liver homogenates [33]. This FOH kinase activity was located in rough and smooth microsomes and associated with the inner, luminal surface of the vesicles. Further analysis identified an activity capable of phosphorylating FP to FPP. Although the biological function of this pathway remains poorly understood, it is likely to be a mechanism for regulating or bypassing the isoprenoid biosynthetic pathway, recycling isoprenoids released from prenylated metabolite degradation, or even facilitating the use of exogenous isoprenoids [32,34–37]. Biochemical evidence suggest that the pathway is carried out by two separate enzymes: a CTP-dependant prenol kinase (PolK) with FOH/GGOH kinase activities that produce FP or GGP and a polyprenyl-phosphate kinase. However, only a few genes encoding these enzymes have been experimentally identified in plants, including the FOLK gene which encodes a FOH kinase (FolK) in *Arabidopsis thaliana* [36], the VTE5 gene which encodes a kinase of phytol (a hydrogenated product of GGOH typical from plants) [35], and the gene VTE6 which encodes a phytyl-P kinase [37]. The enzymes responsible for the salvage pathway of prenols in animals and other organisms remain unidentified. The interest in their identification is growing, as recent studies show that prenols can be enzymatically produced by mammal phosphatases [38,39] or metabolized from dietary sources, possibly playing a role in several diseases [40, 41].

To address these issues, our efforts focused on identifying the enzymes responsible for the FOH/GGOH salvage pathway in *P. falciparum*. We identified a gene in malaria parasites that encodes a PolK, experimentally validated for its FOH/GGOH kinase activities. Additionally, using bioinformatics approaches and the creation of conditional gene expression knockout parasites, we sought to understand the biological significance of the FOH/GGOH salvage pathway in malaria parasites.

## 2. Results

### 2.1 Candidates for apicomplexan prenol kinases are homologous to their plant/algae counterparts and belong to a diverse family with multiple gene duplications

We recently demonstrated the ability of parasites to phosphorylate FOH and GGOH [20]. Hence, we started a bioinformatic search for gene candidates to encode a kinase of prenols. As seeds, we used the *A. thaliana* and *Synechocystis* spp. (strain PCC 6803 / Kazusa) phytol kinases (PhyK—encoded by VTE5 genes, UniProt: Q9LZ76 and P74653, respectively) and *A. thaliana*'s FolK (UniProt: Q67ZM7) since their enzymatic activity was already described in the literature [35,36]. Both sequences were defined as control representatives and were used as queries to a survey for homologous sequences in the *P. falciparum* genome in the PlasmoDB database (https://plasmodb.org/). Sequence homology searches were performed using the BlastP algorithm against the protein database of all *Plasmodium* species available (S3 Fig). As a result, only two possible orthologous proteins were obtained in *Plasmodium yoelli* genome (PY17X_1224100 and PYYM_1223600). The ortholog of these genes in *P. falciparum* 3D7 and NF54 strains were further identified (PF3D7_0710300 and PfNF54_070015200, with 100% identity between then) and used in multiple alignments with the control representative sequences and other four sequences with putative annotation for PhyK (S2 Fig). PfNF54_070015200 has 20.67%, 20.83% and 22.16% identity with *A. thaliana* VTE5, *Synechocystis* spp. VTE5, and *A. thaliana*'s FolK, respectively (identity calculated as a percentage considering identical amino acids between sequences, data from UniProt). Phylogenetic analysis of the retrieved representative prenol binding proteins (Figs 2A and S3) display several diverse clades with multiple gene duplications. It is important to remember that this phylogenetic tree aims to explain the gene family evolution and, therefore, might not always follow a species tree. Since this gene family is so widespread among Eukaryote groups, we assume its common ancestral would be as well as old, taken this together with fast evolving genes and multiple events of duplication and loss of genes, one has to take the conclusions of this study with a grain of salt, and we are limiting ourselves to approach the relationships closer to Apicomplexa.

In this sense, we chose the root-external group of dolichol kinase (DolK) and DolK-like containing representatives of each Eukaryote group (Fig 2A), which directly separate the other genes into a monophyletic group related to PhyK/FolK enzymes. The discussed groups are highlighted by coloured boxes as follows: a large group with two copies of each Embriphyota (or Euviridiplantae dark green), PhyK/FolK enzymes from unicellular algae (in green light green), a monophyletic group composed by Dinoflagelata's (yellow), Stramenopila (turquoise) and the specific Apicomplexa monophyletic clade arises (in red). The specific Apicomplexa monophyletic clade contains representative sequences of *Plasmodium*, *Toxoplasma*, *Cryptosporidium*, *Eimeria* and *Babesia* (in red, S4 Fig), with no gene loses among the great Apicomplexan groups, which suggests conservation. Within it a small clade with Dinoflagellate (in yellow) arises, which could be explained by fast evolution or perhaps by an horizontal transfer. Overall, this supports the idea that *Plasmodium*'s PolK is more similar to other Alveolates' proteins. Interestingly, Blast searches also yielded hits for other organisms, such as *Chromera* (Fig 2A).

We assessed the potential of an AlphaFold-derived structural model of the putative *Pf*PolK to bind prenols using a combination of docking and long Molecular Dynamics (MD) simulations. The generated model displays eight conserved transmembrane helices (TM1-8, Fig 2B), a potential CTP binding pocket and possibly a prenol binding pocket (in grey). The putative CTP binding pocket ends in the charged clamp motif (Arg29, Lys30 and His33), whose

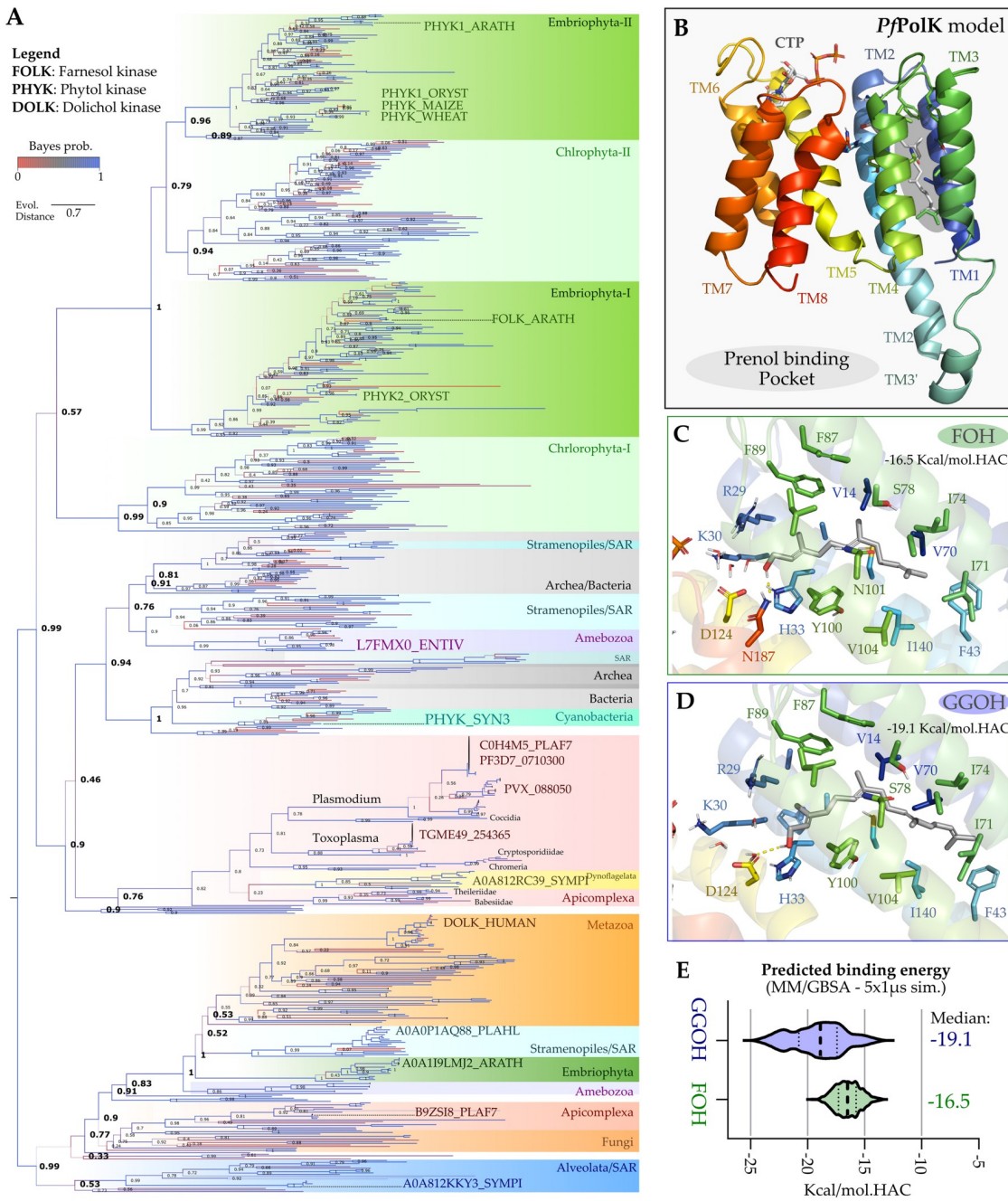

**Fig 2. *Pf*PolK phylogenetic analysis and structural model.** A) Overall phylogenetic dendrogram of prenol binding proteins generated using maximum likelihood method (see Methods). Branch support values (SH2-like) are displayed as numbers for the most relevant clade separation, as well as colours (from the highest scores, in blue, to the lowest values, in red) and thickness of the branches. Root-external group of DolK and DolK-like containing representatives of each Eukaryote super groups, and the other group of PhyK/FolK enzymes. The discussed groups are highlighted by coloured boxes as follows: a large group with two copies of each Euviriplantae (dark green), PhyK/FolK enzymes from unicellular algae (in green light green), a monophyletic group composed by Dinoflagelata's (yellow), Stramenopila (turquoise) and the specific Apicomplexa monophyletic clade arises (in red). Supporting information provides the full phylogenetic tree with all values for branch support and labelled taxa, as well as a key-taxa conversion table. B) AlphaFold 2 *Pf*PolK model displays eight conserved transmembrane helices (TM1-8, coloured) and a potential prenol binding pocket, depicting in the bottom the nucleotide-binding site with the Thre144 and Glu179 composing a hinge region. This model was used to generate the potential binding mode for FOH (green, C) and GGOH (blue, D) by a combination of flexible docking and long molecular dynamics simulations (5x1 µs for each system in explicit solvent and membrane), E) the MD trajectories were utilized to infer the substrates predicted binding energy (Kcal/mol.HAC, where HAC—heavy atom count), suggesting from the median values of the violin-plot displayed distribution that GGOH would have a lower potential binding energy. Dotted lines describe the first quartile amplitude.

positive charges could be used to orient substrate phosphates into the catalytic conformation, while the nucleotide ring appears to be stabilized by a conserved "hinge" region composed by the main chain of Thr144 and the side chain of Glu179 (Fig 2B, down inset). Meanwhile, the potential prenol binding pocket, composed by the TM's 1–4, may accommodate both FOH (Fig 2C) and GGOH (Fig 2D) relying on the conformational change, upon simulation, of Phe43 and Ile71 to fit the latter. The hydroxyl group of both substrates possibly coordinates the Arg-Lys-His triad by conserved water interactions. The MD trajectories were further utilized to infer the substrates predicted binding energy (Fig 2E), suggesting that GGOH (-19.1 kcal/mol) would have a lower potential binding energy, when compared to the FOH (-16.5 kcal/mol), with both substrates being predicted to be able to bind *Pf*PolK.

## 2.2 Farnesol/geranylgeraniol kinase activity of *Pf*PolK

To study the catalytic activity of *Pf*PolK candidate, we expressed its gene heterologously in yeast. This expression system was chosen due to its advantages as a eukaryotic protein-expression system, and because it was previously demonstrated that this organism did not phosphorylate FOH and GGOH [36]. Therefore, the W303-1A strain of *S. cerevisiae* was transformed with p416-GPD vector (empty vector employed, as a control) or engineered to express *Pf*PolK from p416-*Pf*PolK plasmid. To enhance the likelihood of actively expressing *Pf*PolK, no tags were added, and the *Pf*PolK gene sequence was optimized for recombinant expression. Transformant yeasts were grown in an SD-uracil drop-out medium and employed for enzymatic assays. The incubation of yeast extracts with [$^3$H] FOH or [$^3$H] GGOH plus CTP produced radiolabeled compounds chromatographically compatible with their respective phosphates. The formation of polyprenyl phosphates was not observed in assays without the addition of CTP or employing wild-type yeasts transformed with the empty vector (Fig 3). No compounds displaying chromatographic compatibility with FPP and GGPP were detected. Although we used the same amounts of substrates in all assays, the chromatographic spots compatible with FOH and FP were more visible than those compatible with GGOH and GGP. This issue is likely caused by varying extraction efficiencies between different prenols.

## 2.3 Conditional knockout uncovers the relationship between P*f*PolK and MEP inhibitors

After confirming the catalytic activity of *Pf*PolK, we investigated its biological importance in malaria parasites by generating conditional knockout NF54 parasites for PfNF54_070015200 (Fig 4A and 4B). Clones PolK-loxP-C3, E3, and G9 were obtained and the presence of the recodonized PolK was assessed by PCR employing the primers PolK-recod-forward and PolK-recod-reverse (Fig 4C); genomic integration was also confirmed by PCR employing the primers Int-control-forward and Int-recod-reverse (Fig 4D).

Expected excision of the floxed *Pf*PolK-loxP was also confirmed by PCR using the primers PolK-HR1-forward and PolK-HR2-reverse at 24, 48, 96 and 144 h after knockout induction with rapamycin, generating Δ-PolK parasite lines (Fig 4E). Importantly, Δ-PolK parasites showed no reduction in growth and could be maintained indefinitely in culture, as demonstrated by monitoring their growth for up to 144 hours (Fig 4F). Western blots revealed that transfected parasites had an HA epitope fused to a ∼31 kDa protein, matching the expected size of the PolK protein. By contrast, HA-tagged PolK protein was not detectable in Δ-PolK parasites 48 hours after induction of *Pf*PolK-loxp excision with rapamycin (Fig 4G; see original images and other replicates in supporting information, S5A and S5B Fig). Immunofluorescence assays also revealed that the HA-tagged protein did not exclusively co-localize with the apicoplast (Fig 4H and 4I; other replicates in supporting information, S5C Fig), suggesting

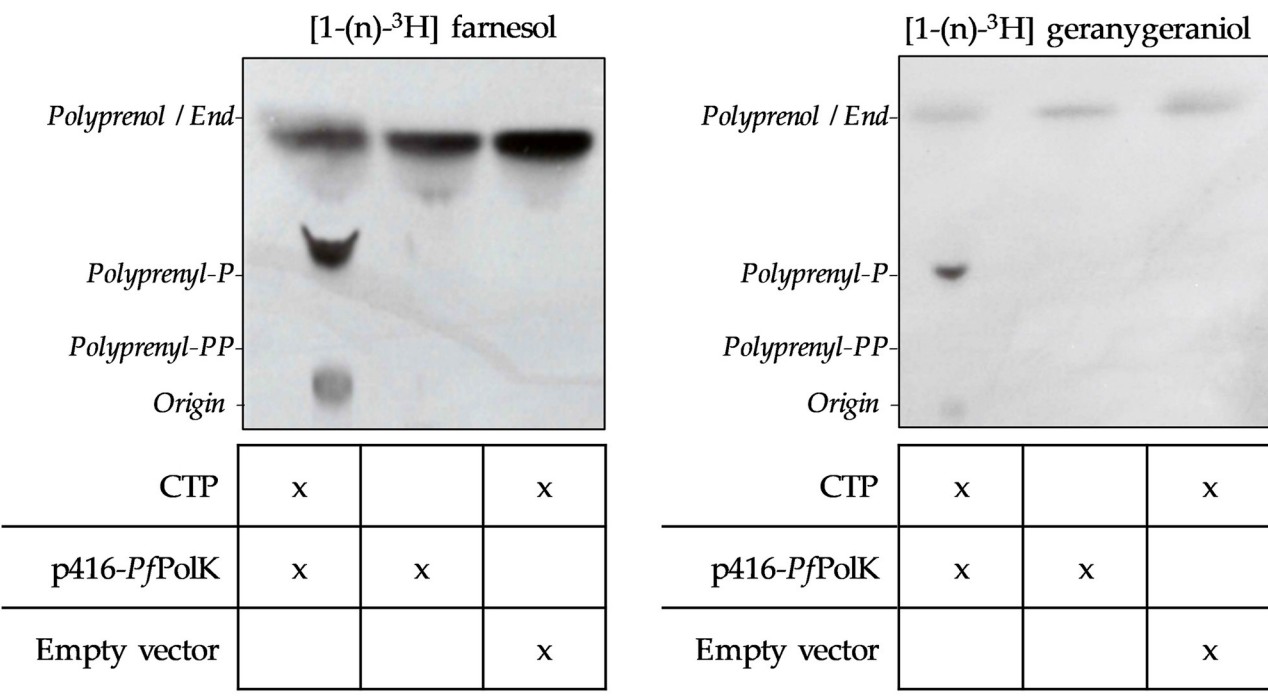

**Fig 3. Farnesol and geranylgeraniol kinase activities.** Autoradiographs of the PolK enzymatic activity assays using [³H] FOH or [³H] GGOH as substrates and chromatographed by TLC. The enzyme source of these assays came from whole extracts of yeast strains transformed with either the empty vector (p416-GPD) or p416-*Pf*PolK. Compounds added to the enzymatic reaction are indicated under the TLC autoradiography image. The retention of different standards is also indicated. These experiments were repeated three times with similar results.

that the prenol salvage pathway could be at least partially, apicoplast-independent. Previous studies indicated the gene's essential role in *P. falciparum* [42], while research on rodent malaria parasites suggested its dispensability [43]. The fact that Δ-PolK exhibit no changes in growth rates is consistent with the findings in rodent parasites. These results also indicate that, although the FOH/GGOH salvage pathway might serve as a significant alternative source of isoprenoids, it may not be critical for parasite survival, likely due to the parasite's capacity for endogenous isoprenoid biosynthesis via the MEP pathway.

The susceptibility of these parasites to MEP inhibitors was studied next. Loss of the PolK gene increased the sensitivity of mutant parasites to fosmidomycin two-fold, when compared to wild type parasites ($IC_{50}$ fosmidomycin 1.09 ± 0.33 μM vs 0.51 ± 0.07 μM) (Fig 5A and 5B). No significant differences were observed in the effect of clindamycin under the same conditions, possibly due wider effects on parasite metabolism. As expected, the presence of prenols, FOH or GGOH, in the medium of control parasites reduced their sensitivity to fosmidomycin. This was evidenced by a 3-fold increase in the $IC_{50}$ for fosmidomycin in the presence of FOH, and a 15-fold increase in the presence of GGOH, when compared to controls grown without prenol supplementation. Likewise, the $IC_{50}$ for clindamycin increased 5-fold in the presence of GGOH, as compared to the control group with no-additions (Fig 5C and 5D). Remarkably, the addition of prenols did not have any rescue effects on the antimalarial activity of fosmidomycin or clindamycin in Δ-PolK parasites (see Fig 5A-D). This strongly suggests that the presence of PolK is necessary for the rescue effect of prenols on the antimalarial activity of MEP inhibitors.

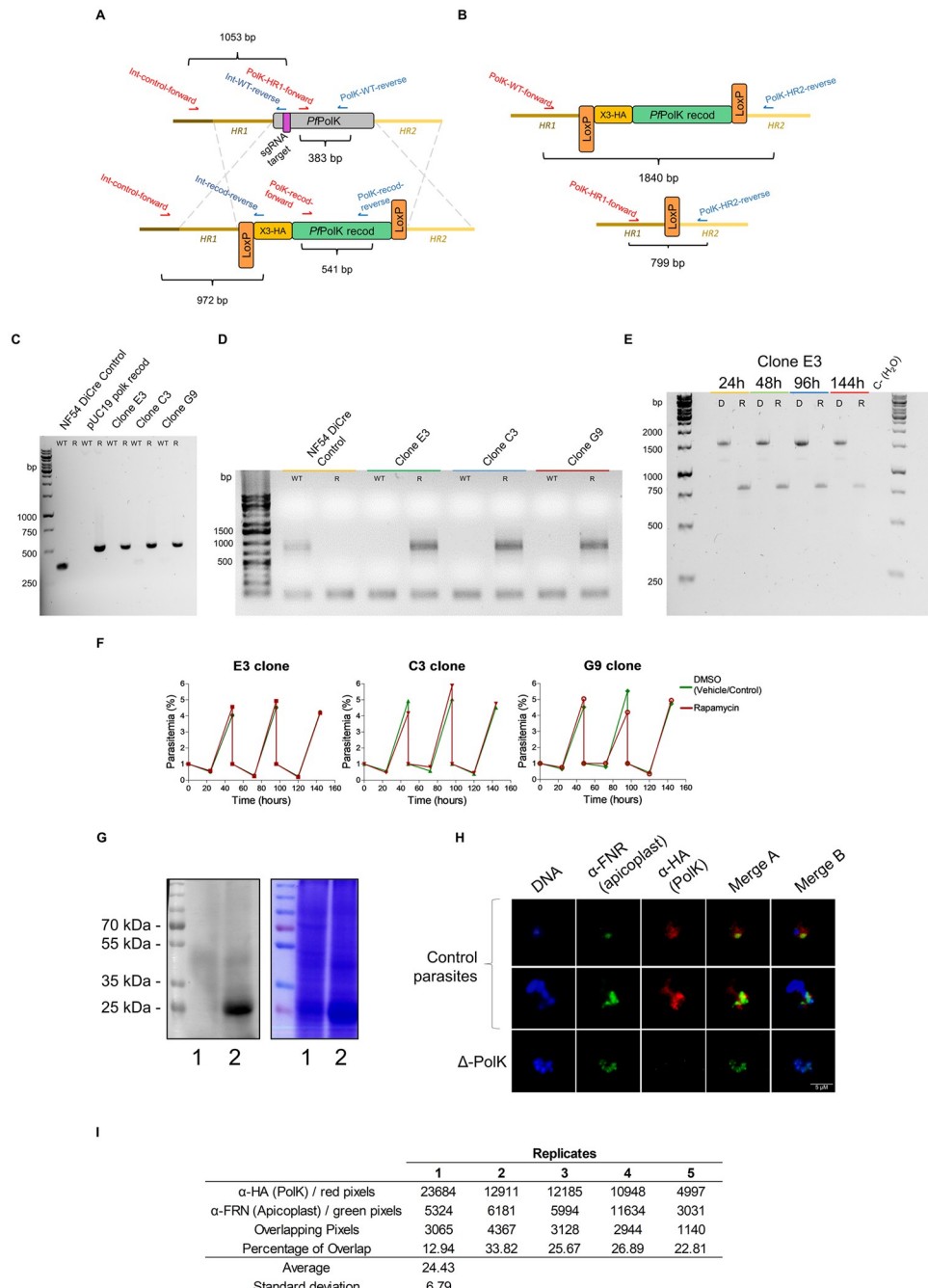

**Fig 4. Conditional knockout of *P. falciparum* PolK gene.** (A) Diagram depicting the edition of single-exon gene *P. falciparum* PolK. Using Cas9-assisted genome editing, all 612 bp of the native *Pf*PolK (WT) open reading frame were replaced by a recodonized sequence (*Pf*PolK recod) and a 3x-HA sequence (yellow box) in the 5' end of the sequence and all flanked by two loxP sites (orange boxes). The position of the 20-nucleotide region targeted by the single guide RNA (sgRNA) is indicated (purple box). (B) Diagram of the rapamycin-induced site-specific excision. Recombination between the loxP sites removes the entire recodonized gene (green box) and 3-HA sequence; (C) PCR detection of transgenic parasites. Wild type DNA (NF54 DiCre control) was confirmed using PolK-WT-forward and reverse primers, amplifying a fragment of 383 bp. Using PolK-recod-forward and reverse primers it was detected a fragment of 541 bp corresponding to the redoconized version of PolK in transgenic parasites (Clones E3, C3 and G9) and in a plasmid containing the recodonized PolK (pUC19 polk recod) as a control. It was used as a standard the GeneRuler 1 kb DNA Ladder (#SM0311, Thermo Scientific). (D) PCR assessment of the genome integration of the construct in the *Pf*PolK locus. Wild type DNA (NF54 DiCre control) was confirmed using Int-control-forward and Int-WT-reverse primers, amplifying a fragment of 1053 bp. Using Int-control-forward and Int-recod-reverse primers it was detected a

fragment of 972 bp corresponding to the integrated locus. It was used as a standard the GeneRuler 1 kb Plus DNA Ladder (#MK-130, Cellco) (E) Confirmation of the rapamycin-induced *Pf*PolK gene excision in the clone PolK-loxP E3. The deletion was confirmed by PCR 24, 48, 96 and 144 h after treatment with DMSO (D) or rapamycin (R) using primers PolK-HR1-forward and PolK-HR2-reverse (in red and blue on panel B, respectively). Excision reduces the amplicon from 1840 bp to 799 bp, disrupting *Pf*PolK. It was used as a standard the GeneRuler 1 kb DNA Ladder (#SM0311, Thermo Scientific). (F) Figure shows the 144h evolution of parasitemia in different clones in which *Pf*PolK gene was excised (parasites exposed to rapamycin) or not (parasites exposed to DMSO). (G) Western blot of transgenic parasites (left) and the respective Comassie stained gel (right). Western blot was performed to analyse the HA-tagged *Pf*PolK of parasites in which *Pf*PolK was excised (lane 1, parasites exposed to rapamycin) or preserved (lane 2, parasites exposed to DMSO). (H) Immunofluorescence analysis of HA-tagged *Pf*PolK of parasites in which *Pf*PolK was excised (parasites exposed to rapamycin, Δ-Polk) or not (control parasites, exposed to DMSO). HA-tagged *Pf*PolK is marked in red, the apicoplast is green and the nucleus in blue. For all experiments, the excision of PolK-loxP parasites was initiated by adding 50 nM rapamycin or DMSO (used as a vehicle control) to synchronized ring-stage cultures. The cells underwent a 24-hour treatment period, followed by a washout step. At this point, we considered it as time zero, and the parasites were cultured again without rapamycin. For Western blot and IF analysis, parasites were utilized at the trophozoite/shizont stage (at 24 hours). In the case of PCR analysis and the assessment of parasitemia evolution, parasites were collected at various time points, as indicated in the figure. (I) Quantitative pixel analysis of immunofluorescence images. The table presents data from the analysis of pixel overlap between the red (anti-HA antibody staining) and green (anti-FRN antibody staining) channels in five individual parasites. The count of red, green, and overlapping pixels was performed using the ImageJ software. The "Percentage of Overlap" was calculated as the ratio of overlapping pixels to the total red channel pixels, multiplied by 100, for each parasite replicate. The average and standard deviation (SD) of the percentage overlap across the five replicates are also provided.

## 2.4 Farnesol and geranylgeraniol phosphorylation is required for their utilization for protein prenylation

As mentioned above, lack of protein prenylation disrupts the function of the digestive vacuole and leads to the loss of parasitic homeostasis [18]. To understand the effects of PolK deletion on FOH utilization, the incorporation of [$^3$H] FOH and [$^{14}$C] isoleucine (control) into proteins was determined in parasites with functional *Pf*PolK and after the excision of the gene (Fig 6). The deletion of *Pf*PolK resulted in a significant decrease in counts per minute (CPM) corresponding to [$^3$H] FOH-labeled proteins compared to parasites with an intact PolK enzyme. Only a few counts were still detected in Δ-PolK parasites, likely corresponding to the remaining dolichol-P oligosaccharide and/or other radiolabeled lipids that were not covalently bound to proteins. It is worth noting that all parasites incorporated similar levels of [$^{14}$C] isoleucine into proteins, indicating no observable defects in protein synthesis other than prenylation.

## 2.5 GGOH is present in human plasma and media supplements

The higher susceptibility of Δ-PolK parasites to fosmidomycin (Fig 5A) could be attributed to the presence of prenols in the culture media and their inability to salvage them. To clarify this, the unsaponifiable lipids from the AlbuMAX I lipid-rich media supplement were analysed using triple quadrupole GC-MS. Additionally, we examined a human plasma sample, another common standard supplement in *P. falciparum* culture media. GGOH was detected in both samples (Fig 7A), while FOH was not present or at concentrations below the detection limit of our method. These results show, for the first time, that GGOH is present both in AlbuMAX I and human plasma [44], although further studies will be required to evaluate the concentration range of GGOH in human plasma. We also investigated whether prenol levels present in human plasma could impact fosmidomycin efficacy in cultured parasites. Thus, we compared fosmidomycin IC$_{50}$ values in media supplemented with 10% whole or delipidated human plasma, from different donors, in control and *Pf*PolK excised parasites (Fig 7B). Despite inherent variability, which could be attributed to differences in plasma composition among donors,

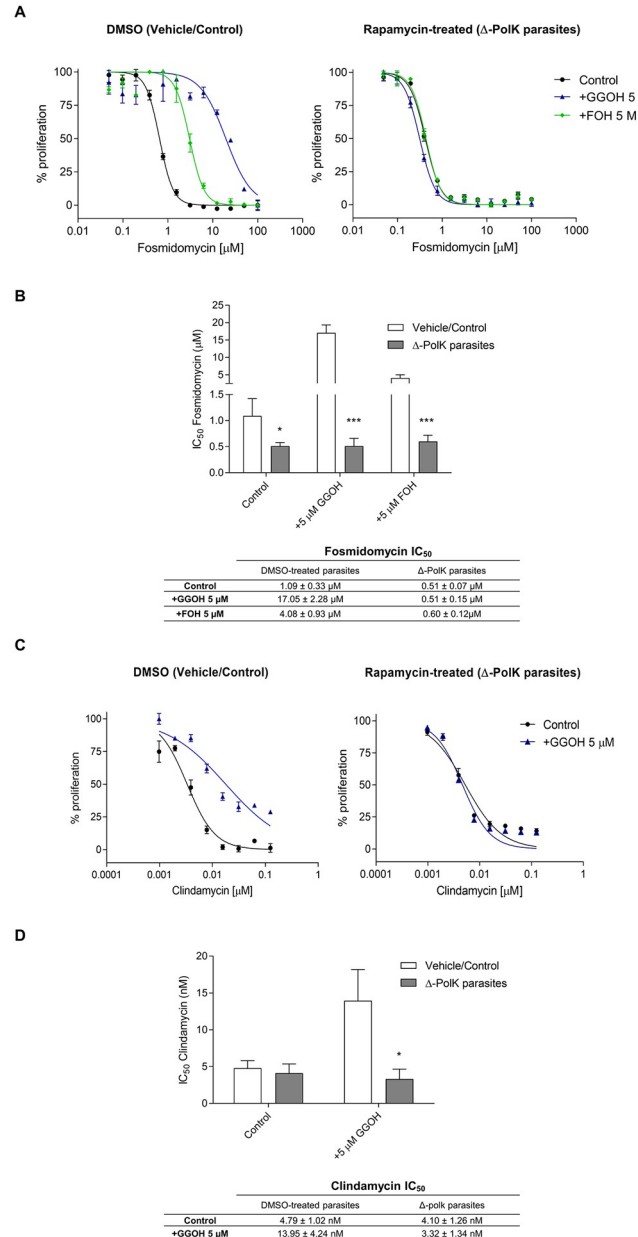

**Fig 5. Phenotypic characterization of knockout parasites.** (A) Fosmidomycin dose-response curves after 48h in parasites maintaining a functional *Pf*PolK (DMSO (Vehicle/control)) or Δ-PolK parasites. These parasites were cultured in RPMI medium in the presence or absence of the indicated prenols (5 μM). (B) Fosmidomycin IC$_{50}$ values of the results exposed in the previous panel. (C) Clindamycin dose-response curves after 96 h in parasites maintaining a functional *Pf*PolK (DMSO—Vehicle/control) or Δ-PolK parasites. These parasites were cultured in RPMI medium in the presence or absence of GGOH (5 μM), as indicated. (D) Clindamycin IC$_{50}$ values of the results exposed in the previous panel. Statistical analysis was made using one-way ANOVA/Dunnet's Multiple Comparison Test.*p<0.05, **p<0.01, ***p<0.001. Comparison made to Vehicle/Control data. Error bars represent standard deviation (n = 3). The excision of PolK-loxP parasites was initiated one parasitic cycle before the start of the experiments. This was achieved by adding 50 nM rapamycin or DMSO (used as a vehicle control) to synchronized cultures in the ring stage. These cells underwent a 24-hour treatment period, followed by a washout step, and then used for experiments. Parasitic growth in all experiments depicted in this figure was monitored by flow cytometry.

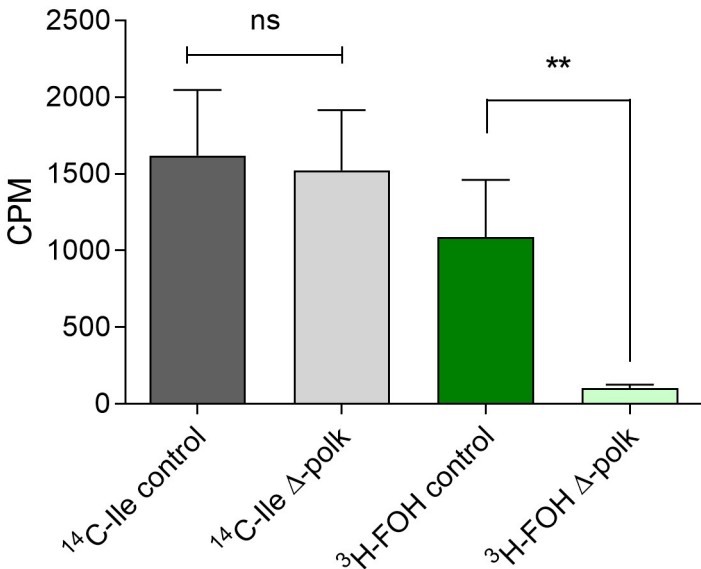

**Fig 6. Farnesol and isoleucine incorporation into proteins in Δ-PolK parasites.** The graph shows the levels of incorporation of $^3$H-FOH and $^{14}$C -isoleucine ($^{14}$C -Ile) into TCA-precipitated proteins in parasites maintaining or not a functional *Pf*PolK. Statistical analysis was made using one-way ANOVA One-way ANOVA / Tukey's Multiple Comparison Test.*p<0.05, **p<0.01, ***p<0.001. Comparison made to between samples of parasites exposed to the same radiolabelled precursor but maintaining or not *Pf*PolK. Error bars represent standard deviation (n = 3). The excision of PolK-loxP parasites was initiated one parasitic cycle before the start of the experiments. This was achieved by adding 50 nM rapamycin or DMSO (used as a vehicle control) to synchronized cultures in the ring stage. These cells underwent a 24-hour treatment period, followed by a washout step, and then used for experiments.

media supplementation with either plasma or delipidated plasma consistently resulted in higher fosmidomycin sensitivity for Δ-PolK parasites.

On average, there was an 11-fold increase in sensitivity in *Pf*PolK excised parasites compared to control parasites. Control parasites cultivated in media with delipidated plasma exhibited greater sensitivity to fosmidomycin, with an average threefold increase compared to those grown in medium supplemented with whole plasma. In contrast, Δ-PolK parasites showed a similar sensitivity to fosmidomycin in media supplemented with whole plasma or delipidated plasma (Fig 7C). These results strongly suggest that human plasma might contain factors that reduce the efficacy of fosmidomycin. Moreover, these factors are partially removed by plasma delipidation techniques, and require the PolK gene for their utilization. The incomplete removal of fosmidomycin-rescue factors through delipidation may be attributed to either shortcomings in the delipidation technique or to the natural occurrence of these factors in the erythrocytes used for parasite culture.

## 3. Discussion

Malaria parasites can grow indefinitely in the presence of fosmidomycin or ribosome inhibitors if exogenous IPP is added to the culture medium [16]. Moreover, pharmacologically-induced isoprenoid biosynthesis deficiency can be transiently mitigated by the addition of FOH, GGOH and unsaponifiable lipids from food, such as sunflower oil and arugula [16–18,20]. In a recent study, our group showed that radiolabelled FOH and GGOH can be incorporated into various long-chain prenols (>20C in length), dolichols, and proteins [20]. However, since all characterized polyprenyl transferases and synthases use polyprenyl pyrophosphates as their natural substrates, and there is no evidence that these enzymes can

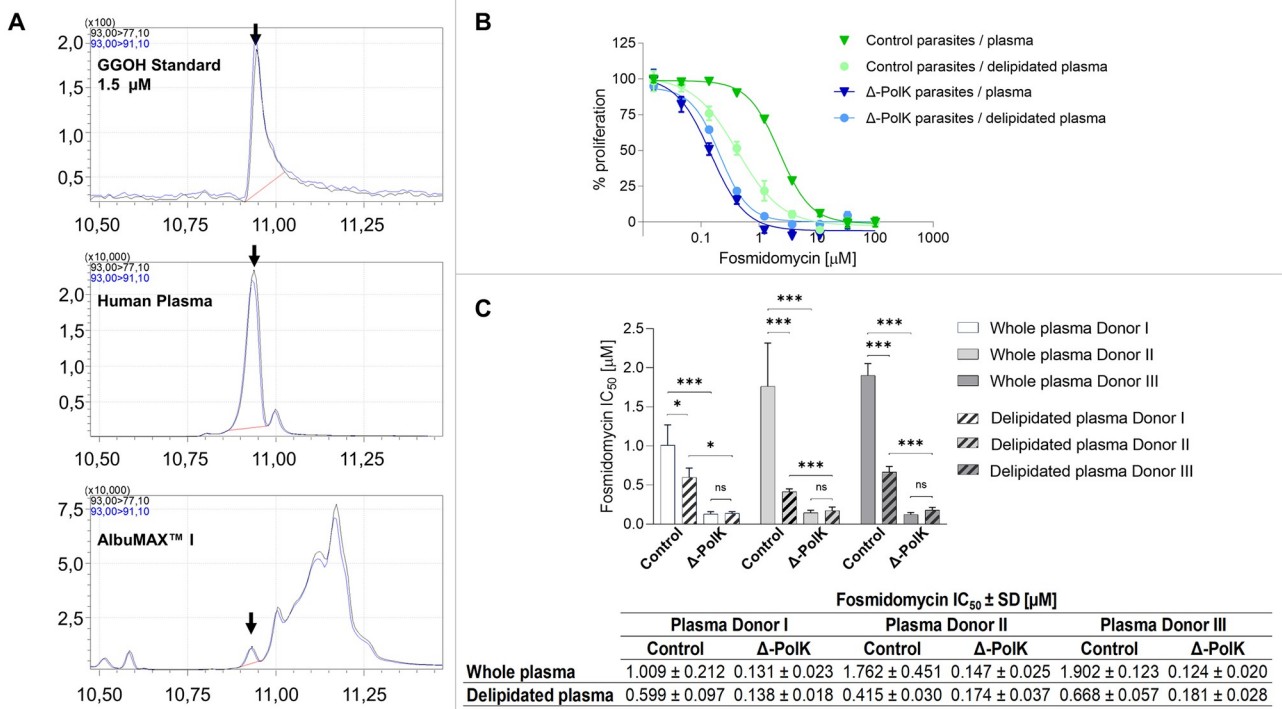

**Fig 7. Sensitivity to fosmidomycin using human plasma.** (A) The images display mass spectrometry obtained using GC-MS triple quadrupole MRM analysis of the standard of GGOH, as well as GGOH detected in human plasma and AlbuMax I. GGOH is highlighted with arrows. (B) A representative image displaying the sigmoidal dose-response effect of fosmidomycin on cultures of either control or Δ-PolK parasites grown in media supplemented with either whole plasma or delipidated plasma from donor II. (C) The image presents bar graphs and a table displaying the $IC_{50}$ values of the antiplasmodial effect of fosmidomycin on Δ-PolK parasite cultures supplemented with either whole or delipidated plasma from three distinct donors. The analyses in (B) and (C) were conducted on parasites subjected to rapamycin-induced knockout or those retaining *Pf*PolK function. The excision of PolK-loxP parasites commenced one parasitic cycle prior to the initiation of the experiments, achieved by adding 50 nM rapamycin or DMSO (serving as a vehicle control) to synchronized ring-stage cultures. These cells underwent a 24-hour treatment, followed by a washout phase, and were utilized for experiments the subsequent day. Parasite growth in all the experiments shown in this figure was assessed via SYBR Green I DNA staining at 72 hours. Statistical evaluations employed one-way ANOVA followed by Dunnett's Multiple Comparison Test. *$p<0.05$, **$p<0.01$, ***$p<0.001$ when compared to Vehicle/Control data. Error bars illustrate standard deviation (n = 3).

use prenols, a phosphorylation pathway for FOH and GGOH salvage is required in malaria parasites. Therefore, we hypothesized the existence of a phosphorylation pathway for FOH and GGOH salvage in malaria parasites and, remarkably, we recently described that *P. falciparum* phosphorylate [3H] FOH and [3H] GGOH into their pyrophosphate counterparts. These results significantly support the existence of a plasmodial FOH/GGOH salvage pathway [20]. In photosynthetic organisms the phosphorylation of FOH and GGOH is carried out by two separate enzymes: a prenol kinase and a prenyl-phosphate kinase [20]. However, only a few genes have been unequivocally identified to encode those enzymes [35–37]. Despite the scarce literature on prenol kinases, we were able to identify a candidate for PolK in *P. falciparum* through BLAST analysis. This enzyme was heterologously expressed in *S. cerevisiae* and its FOH and GGOH kinase activity was biochemically confirmed. Remarkably, *Pf*PolK only catalyses lipid mono-phosphorylations, although it can use either FOH or GGOH as lipid substrates. Consequently, the parasite probably possesses at least another enzyme with prenyl-P kinase activity to make prenols available for further use. The low sequence similarity shared by these enzymes is probably the reason for not finding it in the initial searches.

The results here presented could represent a significant contribution to the field of isoprenoid metabolism, as it links the plasmodial prenol kinase function to a new class of enzymes in

nature. Remarkably, this is the first FOH/GGOH kinase enzyme discovered in a non-photosynthetic organism. As observed in the phylogenetic analysis, putative prenol kinases comprise a very divergent group of clades. This finding supports the idea that *Plasmodium*'s PolK is more similar to unicellular algae proteins, consistent with the endosymbiosis event that occurred in apicomplexan ancestors. Also, the paucity of verified enzymes in all commented clades open the possibility to assign pyrophosphorylating activity to some of these putative enzymes. The model provided possible structural insights into *Pf*PolK such as eight conserved transmembrane helices and putative CTP binding pocket that could be used to orient the phosphates into the catalytic conformation. Additionally, the nucleotide ring is stabilized by a putative conserved hinge region. The model also indicated a possible prenol binding pocket composed of TM's 1–4, possibly capable to accommodate both FOH and GGOH. Despite these remarks, it should be noted that all these computational predictions need further experimental validation.

After *Pf*PolK identification, we studied the biological relevance of the FOH/GGOH salvage pathway. *Pf*PolK appears to be expressed throughout all stages of asexual intraerythrocytic development [45], highlighting its potential importance in the parasite's lifecycle. Previous studies employing Piggybac mutagenesis indicated that the gene is essential in *P. falciparum* [42], whereas studies in rodent malaria parasites predicted it to be dispensable [43]. Interestingly, Δ-PolK parasites remained fully viable with no observable growth rate changes, in agreement with the observations made in rodent parasites. We suggest that the failure of Piggybac mutagenesis for *Pf*PolK may be due to its small size. These results suggest that while the FOH/GGOH salvage pathway is an important source of isoprenoids if prenols are exogenously added to culture, it is not essential for parasite survival, probably due to the endogenous biosynthesis of isoprenoids via the MEP pathway. IFAs revealed that *Pf*PolK did not exclusively co-localizes with the apicoplast marker (Figs 4G and S5C), suggesting that the prenol salvage pathway is at least partially, apicoplast-independent. This contradicts previous large-scale studies which target this protein to the apicoplast [46]. In line with our findings, bioinformatic deep learning signal prediction indicates that this enzyme is localized in the endoplasmic reticulum of malaria parasites [47]. In fact, other studies indicate that the majority of prenol kinase activities in animals and bacteria are located in the approximately 10,000 x *g* supernatant of tissue homogenates, rough and smooth microsomes, and associated with the inner, luminal surface of the vesicles [32].

Δ-PolK parasites displayed greater sensitivity to fosmidomycin and could not be rescued from the antimalarial effects of fosmidomycin or clindamycin by the addition of FOH or GGOH. As far as we know, this observation is the first evidence of FOH and GGOH phosphorylation as a mandatory step for their utilization by living organisms, highlighting the biological relevance of the FOH/GGOH salvage pathway particularly when de novo isoprenoid biosynthesis is inhibited. The biological relevance of the FOH/GGOH salvage pathway in *P. falciparum* raises important questions. In our view, this pathway could serve as a complementary alternative source of isoprenoids for the parasite, independent of the Apicoplast. Furthermore, the pathway could be important for recycling prenols released from the degradation of endogenous prenylated metabolites. This mechanism is similar to the prenol salvage pathway in plants, which also recycles prenols from chlorophyll degradation [34,35]. In addition, the *P. falciparum* FOH/GGOH salvage pathway in the parasite may serve as a mechanism to scavenge some FOH and GGOH derived from the host. Thus, prenol scavenging could help the parasite optimize its energy usage for isoprenoid biosynthesis and partially complement its metabolic requirements. Contrary to this hypothesis, other authors previously concluded that the parasite may use exclusively endogenous isoprenoid biosynthesis based on the observation that increasing concentrations of human plasma components in culture did not affect the

antimalarial effect of a ribosomal inhibitor [16]. However, our experiments suggest that human plasma and AlbuMAX I may contain components that can diminish the effectiveness of fosmidomycin, probably prenols. Considering that in a natural environment, infected erythrocytes are exposed to pure plasma, it is noteworthy that the concentration of these factors in plasma remains sufficient to influence fosmidomycin's efficacy, even when the plasma is diluted tenfold with RPMI medium—this represents the maximum plasma concentration used in culture. Furthermore, GC/MS analysis of human plasma and AlbuMAX I detected GGOH. Contrarily, FOH was not detected, which might either not be present in blood or be below the limit of detection by our methods. The observation that both parasites preserving PolK and Δ-PolK exhibit a more similar sensitivity to fosmidomycin under delipidated plasma-supplemented medium indicates that this rescue phenomenon is most likely due to prenols rather than other substances. Despite these novel results, several aspects remain unknown for now, such as the concentration of GGOH in plasma and whether its origin is the animal isoprenoid biosynthesis pathway and/or dietary sources [40]. Considering that the physiology of prenols' oral absorption and excretion in animals is largely unknown, it's possible that blood concentrations of these substances can vary significantly over time [40,41]. Considering this, we believe that further studies should be carried out to determine if, in clinical settings, the FOH/GGOH salvage pathway scavenges host prenols and might reduce fosmidomycin's efficacy against malaria parasites. The identification of *Pf*PolK opens up new avenues for conducting these studies and may ultimately lead to a better understanding of isoprenoid metabolism in malaria parasites.

Besides malaria parasites, the dietary consumption of GGOH found in foods like vegetable oils has already been shown to have implications for cancer therapy [40]. This study also reveals that dietary supplementation of GGOH in animal models may limit the efficacy of statins (MVA pathway inhibitors) in treating ovarian cancer [40]. Furthermore, GGOH-rich food extracts demostrated to block statin-induced regression of ovarian cancerous cells *in vitro* [40]. However, it should be noted that de Wolf *et al.* [40] performed supplementation assays, which may lead to unphysiological concentrations of GGOH in mice, and thus, further research is required to confirm whether this occurs naturally. Nevertheless, these findings demonstrate that dietary prenols are metabolized by mice and warrant further exploration to determine if their consumption may have a significant impact on cancer therapy. The use of isoprenoid biosynthesis inhibitors has also been explored in other MEP pathway-dependent pathogens such as *Toxoplasma* [48], *Babesia* [49], and *Mycobacterium tuberculosis* [50], as well as in MVA pathway-dependent parasites such as *Leishmania* [51]. Importantly, most of these pathogens can also be rescued from fosmidomycin by FOH and GGOH [18,27,49,52] and thus, possibly possess an active prenol salvage pathway. Hence, we believe it is important to communicate the discovery of PolK so that the scientific community can explore its potential role in other organisms. For example, it has already been demonstrated that *Toxoplasma gondii* relies on both isoprenoid biosynthesis and host intracellular isoprenoids [52], a phenomenon which limits the efficacy of isoprenoid inhibitors against this parasitic infection. The identification and characterization of *Pf*PolK as an enzyme critical for the FOH/GGOH salvage pathway not only provide new insights into the biochemical mechanisms of malaria parasites but also open up new avenues for the study of several other infectious and non-infectious diseases related to isoprenoid metabolism.

## 4. Conclusions

The focus of this work was the identification of the enzymes responsible for the salvage pathway of FOH and GGOH in the parasite and their relationship with MEP-targeting drugs. As a

result, we identified *Pf*PolK, a novel lipid kinase. Through biochemical and molecular approaches, the catalytic activity and biological importance of this transmembrane enzyme were characterized. Our data revealed the non-essential role of *Pf*PolK in parasite survival and its crucial involvement in the use of exogenous prenols for protein prenylation. *Pf*PolK is also key for maintaining cell homeostasis under the effects of MEP inhibitors. Indeed, we think the findings of this study are relevant to understand the fascinating metabolism of malaria parasites.

## 5. Materials and methods

### 5.1 Reagents, stock solutions and parasitic strains

AlbuMAX I Lipid-Rich BSA and RPMI-1640 were purchased from Thermo Fisher Scientific (Leicestershire, UK). Dolichol and dolichyl-P 13–21 were purchased from Avanti (Alabama, USA). [1-(n)-$^3$H] GGOH (14 Ci/mmol; 1 mCi/mL), [1-(n)-$^3$H] FOH (14 Ci/mmol; 1mCi/mL) and L-[4,5-$^{14}$C (U)] isoleucine (200–300 mCi/mmol; 0.1 mCi/mL) were purchased from American Radiolabeled Chemicals (St. Louis, USA). SYBR Green I nucleic acid gel stain and SYTO 11 were purchased from Thermo Fisher Scientific (Waltham, Massachusetts, EUA). Sterile stock solutions were prepared at 10 mM for fosmidomycin sodium salt hydrate in water, 2 mM clindamycin hydrochloride in water, 125 mM of GGOH in ethanol and 200 mM of each other non-radiolabelled prenols in ethanol. All other reagents were purchased from Sigma (St. Louis, Missouri USA) or specific companies, as cited in the text. Polyprenyl phosphates were obtained by mild acid treatment of the respective commercial pure pyrophosphates (Sigma; Sigma codes F6892 and G6025) [53]. For this work, a Cre-LoxP *P. falciparum* NF54 strain [54], a generous gift of Moritz Treeck (The Francis Crick Institute, London, United Kingdom), was employed. Erythrocytes and CPDA-1 treated hospital-grade plasma bags for transfusions (AB+) were obtained from the Blood Center of Sírio Libanês Hospital (São Paulo, Brazil).

### 5.2 *P. falciparum in vitro* culture and synchronization

*P. falciparum* NF54 DiCre cells were cultured *in vitro* following the Trager and Jensen culture method employing RPMI-1640 medium completed with 0.5% AlbuMAX I Lipid-Rich BSA. Parasites were maintained in 75 cm$^2$ cell culture flasks at 37°C [55–57]. The culture medium pH was adjusted to 7.4 and was introduced a gas mixture of 5% CO$_2$, 5% O$_2$ and 90% N$_2$ purchased from Air Products Brasil LTDA (São Paulo, SP, Brazil). Parasite synchronization at ring stage was performed with 5% (w/v) D-sorbitol solution as described previously [58]. Parasite development was monitored microscopically on Giemsa-stained smears. PCR for mycoplasma and optic microscopy were used to monitor culture contamination [59].

### 5.3 Metabolic labelling of parasites

Our work focused on biochemical experiments in schizont stages because of previous studies showing a higher incorporation rate of [$^3$H] isoprenic moieties at this stage [60]. For this, synchronous cultures of *P. falciparum* at the ring stage in 25 cm$^2$ flasks were labelled with either 0.75 μCi/ml [$^3$H] FOH or 40 μCi/ml [$^3$H] isoleucine employed as control of protein synthesis [61]. After 12–16 h, parasites at trophozoite/schizont stages were obtained by saponin lysis [62]. For this, cultures pellets were lysed with 30 mL 0.03% saponin in PBS at 4°C. Parasites were then centrifuged at 1,500 *x g* for 5 min at 4°C and subsequently washed in PBS.

## 5.4 Assessment of radiolabelled proteins

The assessment of radiolabeled proteins was performed following a similar protocol as described elsewhere [63]. Radiolabeled parasites were suspended in 100 μL of lysis buffer (2% w/v SDS, 60 mM DTT in 40 mM Tris-Base pH 9). The samples were then cooled at room temperature, and proteins were precipitated by adding 20% trichloroacetic acid (TCA) in acetone at 4˚C. The samples were kept on ice for 5 minutes, and the proteins were collected by centrifugation at $12,000 \times g$ for 10 minutes. The precipitate was washed three times with 80% acetone. Subsequently, the proteins were dissolved by incubating them at 90˚C in alkaline buffer (0.5 M NaOH, 25 mM EDTA, 0.1 w/v SDS in water) for 30 minutes. Finally, 1 mL of liquid scintillation mixture (PerkinElmer Life Sciences, MA, USA) was added to the samples. After vortex, the radioactivity of samples was measured using a Beckman LS 5000 TD β-counter apparatus (Beckman, CA, USA) and results were analysed using GraphPad Prism software.

## 5.5 Drug-rescue assays in malaria parasites

In some cases, the dose-response curve was calculated and the concentration of drug/metabolite required to cause a 50% reduction in parasite growth ($IC_{50}$ value). Assays started at the ring stage at 1% or 0.15% parasitemia and had a duration of 48 h or 96h. Serial dilutions of the antimalarials were prepared in 96-well microplates in RPMI complete medium supplemented or not with FOH / GGOH. Solvent controls and untreated controls were always included and results were analysed by GraphPad Prism software. All experiments which monitor parasitic growth were performed at least three times with three or four technical replicates. The data were adjusted to a dose–response curve to determine the $IC_{50}$ value. Parasitic growth was monitored by either flow cytometry or SYBR green I DNA staining, as indicated in each experiment.

## 5.6 Monitoring of parasitic growth by flow cytometry

Parasitemia was monitored by flow cytometry using the nucleic acid stain SYTO 11 (0.016 μM) (Life Technologies no. S7573) in a BD LSRFortessa machine as previously described [64] or in a BD FACSCalibur machine as previously described [65].

## 5.7 SYBR green I DNA staining

Culture (100 μl) was incubated in a 96-well cell plate in dark at room temperature after adding 100 μl of SYBR green I 2/10,000 (vol/vol) in lysis buffer (20 mM Tris [pH 7.5], 5 mM EDTA, 0.008% saponin [wt/vol], 0.08% Triton X-100 [vol/vol]) [66]. Fluorescence was measured using a POLARstar Omega fluorometer (BMG Labtech, Ortenberg, Germany) with the excitation and emission bands centered at 485 and 530 nm, respectively. The fluorescence values of uninfected erythrocytes were subtracted from the values obtained for infected cells.

## 5.8 Bioinformatics

**5.8.1 Sequence similarity search and phylogenetic tree.** Sequences from model organisms were retrieved from UniProt, using the term 'prenol kinase' as the keyword. Sequences were retrieved from NCBI/GenBank using the Blast tool (with scoring matrix BLOSUM45 for distant similar sequences) with an e-value cut-off of $10^{-5}$ creating a dataset. Additionally, no similar sequences were found in vertebrate genomes. Sequence renaming and editing were performed with in-house Perl scripts. Sequences with less than 30% global similarity or missing the ORF initiation codon were excluded from further analyses. The full dataset was clustered by similarity (70%) using CD-Hit [67] and a set of representative sequences were

selected for global alignment using ClustalO [68] with the standard options. This algorithm often selects single organisms representing a full clade of highly similar sequences, randomly selecting a centroid sequence within the cluster as a representative. Specific sequences in relevant Alveolates and other Eukaryotic unicellular organisms were derived using *Plasmodium*'s PolK and DolK as queries for Blast (tBlastN, Blossum80) searches against scaffolds and contigs from the vEuPathDB (last access in November 2023). Human DolK sequences were used as seeds for protein blast in the UniProt database in order to find representative homologues for the root group.

Maximum likelihood phylogenetic tree was generated using FastTree 2 [69], with SH-2 like values as branch statistical support. The substitution model WAG was selected for calculations, by ProtTest3 [70], based on the highest Bayesian Information Criterion values. All other parameters, except the equilibrium frequencies, were estimated from the dataset. Dendrogram figures were generated using FigTree 1.4.4 (see http://tree.bio.ed.ac.uk/software/figtree/", last access in November 2023).

**5.8.2 AlphaFold model and molecular docking.** *Pf*PolK model was retrieved from AlphaFold database (sequence: PF3D7_0710300, https://alphafold.ebi.ac.uk/entry/C0H4M5) and prepared using the PrepWizard implemented in Maestro 2022v4 with standard options. All substrate ligands for docking were drawn using Maestro and prepared using LigPrep to generate the three-dimensional conformation, adjust the protonation state to physiological pH (7.4), and calculate the partial atomic charges, with the force field OPLS4. Docking studies with the prepared ligands were performed using Glide (Glide V7.7), with the flexible modality of Induced-fit docking [71,72] with extra precision (XP), followed by a side-chain minimization step using Prime. Ligands were docked within a grid around 13 Å from the centroid of the orthosteric pocket, identified using SiteMap (Schrödinger LCC) [73], generating ten poses per ligand. Docking poses were visually inspected, independently from the docking score, and those with the highest number of consistent interactions were selected for simulation.

**5.8.3 Molecular dynamics simulations.** *Pf*PolK model with the different substrates was simulated to clarify which residues contributed to the stability within the binding site. Molecular Dynamics (MD) simulations were carried out using the Desmond engine [74] with the OPLS4 force-field [75]. The simulated system encompassed the protein-ligand/cofactor complex, a predefined water model (TIP3P) [76] as a solvent, counterions ($Na^+$ or $Cl^-$ adjusted to neutralize the overall system charge) and a POPC membrane based in the transmembrane motifs predicted in the model. The system was treated in an orthorhombic box with periodic boundary conditions specifying the shape and the size of the box as 10x10x10 Å distance from the box edges to any atom of the protein. Short-range coulombic interactions were performed using time steps of 1 fs and a cut-off value of 9.0 Å, whereas long-range coulombic interactions were handled using the Smooth Particle Mesh Ewald (PME) method [77]. PolK+GTP+-substrates systems were then subjected to simulations of 100 ns for equilibration purposes, from which the last frame was used to generate new replicas. The equilibrated system underwent at least 1 μs production simulation, in four-five replicas (total of 5 μs per substrate), followed by analysis to characterize the protein-ligand interaction. The results of the simulations, in the form of trajectory and interaction data, are available on the Zenodo repository (code: 10.5281/zenodo.7540985). MD trajectories were visualized, and figures were produced using PyMOL v.2.5.2 (Schrödinger LCC, New York, NY, USA).

Protein-ligand interactions and distances were determined using the Simulation Event Analysis pipeline implemented using the software Maestro 2022v.4 (Schrödinger LCC). The compounds' binding energy was calculated using the Born and surface area continuum solvation (MM/GBSA) model, using Prime [78] and the implemented thermal MM/GBSA script. For the calculations, each 10th frame of MD was used. Finally, root mean square deviation (RMSD)

values of the protein backbone were used to monitor simulation equilibration and protein changes (S1 Fig). The fluctuation (RMSF) by residues was calculated using the initial MD frame as a reference and compared between ligand-bound and apostructure simulations (S2 Fig).

## 5.9 Generation of conditional Δ-*polk* parasites

A single guide RNA (sgRNA) targeting the PolK genomic locus in PfNF54 strain (PfNF54_070015200) was designed with the CHOPCHOP gRNA Design Tool [79]. To generate the plasmid expressing the *Streptococcus pyogenes* Cas9 and the sgRNA, the primers 5′- AAGTATATAATATTGGACATAGAACAATGTCACAAGTTTTAGAGCTAGAA-3′ and 5′- TTCTAGCTCTAAAACTTGTGACATTGTTCTATGTCCAATATTATATACTT-3′ were annealed and ligated into a BbsI-digested pDC2-Cas9-hDHFRyFCU plasmid (a gift from Ellen Knuepfer) [80]. The donor plasmid, a pUC19 plasmid containing the recodonized PfNF54_070015200 gene, was manufactured by GenWiz Gene Synthesis (Azenta, Chelmsford, USA) with the coding sequence for a 3xHA (Human influenza hemagglutinin) tag in the N-terminal part of the PolK (See the recoded part of the transfection plasmid sequence in S6 Fig). In addition, the cassete was flanked with two loxP sequences and two homology regions of 500 bp corresponding to intergenic regions upstream and downstream of PfNF54_070015200 (HR1 and HR2 respectively). Confirmation of the appropriate modification of the PolK gene after transfection was assessed by diagnostic PCR using primers that specifically recognized the wild-type sequence (PolK-WT-forward: 5'-GGATATAG GAGAGGTTTGCCAC-3' and PolK-WT-reverse: 5'-CCTACTATTGCCGCCATTG-3') or the recodonized sequence (PolK-recod-forward: 5'-GCTTCGTATTGTTCGTGATA-3' and PolK-recod-reverse: 5'-CCACCGAACAACTCTAAGAA-3').

Subsequent limiting dilution was performed to generate clones of PolK locus-modified parasites resulting in the isolation of PolK-loxP-C3, PolK-loxP-E3 and PolK-loxP-G9 clones. The modification of the locus was reconfirmed by PCR as previously described. Additionally, the integration of the cassette into genome was assessed by diagnostic PCR using primers that specifically recognized a locus upstream HR1 (Int-control-forward: 5' TGGGGACAAAACAG-CATTAAACT 3') and the 3xHA sequence (Int-recod-reverse: 5' ACGTCGTATGGGTACATCTGC 3'). As a control for this PCR, we employed the primer Int-control-forward and a primer that specifically recognized the wild-type PolK (Int-WT-reverse: 5' TGGGAATTTGTTTACACTGGTCT 3'). Efficiency of the conditional excision of the floxed PolK-loxP clones was initiated by adding 50 nM rapamycin or DMSO (used as a vehicle control) to synchronized ring-stage cultures. The cells underwent a 24-hour treatment period, followed by a washout step. At this point, we considered it as time zero, and the parasites were cultured again without rapamycin. Subsequently, they were collected at different time points as indicated in each experiment. To demonstrate the efficient excision of PolK-loxP, gDNA from the clones were obtained and used in diagnostic PCR with primers annealing in the homology regions HR1 (PolK-HR1-forward: 5'-ATGATATTTACCATAATTTATGGGC-3') and HR2 (PolK-HR2-reverse: 5'-CTGTTTTTTCTCTTTATTTCCTTCTC-3'). The three clones were used in all subsequent experiments.

## 5.10 Immunofluorescence assays

Before the immunofluorescence assays (IFA), the excision of PolK-loxP parasites was initiated by adding 50 nM rapamycin or DMSO (used as a vehicle control) to synchronized ring-stage cultures. The cells underwent a 24-hour treatment period, followed by a washout step and cultured again without rapamycin. At trophozoite/shizont stage (24 hours latter) parasites were collected for analysis. Before image acquisition, a μ-Slide eight-well chamber slides (Ibidi GmbH, Gräfelfing, Germany)

were incubated in a working poly-L-lysine solution (1:10 dilution from stock 0.1%) for 5 minutes at RT. The poly-L-lysine was then removed with suction and the slides were left to dry. In parallel, parasite cultures were washed 3 times with RPMI medium and 150 μL of culture were placed in the pretreated slide. The cultures were then fixed by adding 150 μL of paraformaldehyde 4% in PBS to the slides and incubating them at 37°C for 30 minutes. After washing the cultures once with PBS, the cultures were permeabilized by adding 150 μL of 0.1% Triton-X-100 in PBS and incubating them at room temperature for 15 minutes. The cultures were then washed 3 times with PBS, and were blocked by adding 150 μL of 3% BSA in PBS and incubating them for 30 minutes at room temperature at 400 rpm, orbital agitation. The cultures were then washed 3 times with PBS and 150 μL of primary antibody solution (Rabbit polyclonal anti-ferredoxin-NADP reductase, diluted 1:100 and Rat Anti-HA, diluted 1:20 in 0.75% BSA/PBS) was then added, and the cultures were incubated overnight at 4°C with at 400 rpm. Afterwards, the cells were washed 3 times with PBS to remove the excess primary antibody solution. The supernatant was removed and secondary antibody solution (Goat anti-Rabbit IgG (H+L) Alexa Fluor 488, #A11034, (Life Technologies, Carlsbad, California, EUA) and Goat anti-Rat IgG (H&L)—AlexaFluor 594, #A11007 (Invitrogen, Waltham, Massachusetts, EUA), diluted 1:100 in 0.75% BSA/PBS, was added to the cultures and incubated for 1 hour at room temperature at 400 rpm, orbital agitation. Hoechst 33342 (Thermo Fisher Scientific, Waltham, Massachusetts, EUA) was diluted 1:1000 in the mix and also added to cultures. The cells were then washed 3 times with PBS to remove excess secondary antibody. The supernatant was removed and the slides maintained in 150 μL of PBS. For microscopy analysis, an Olympus IX51 inverted system microscope, equipped with an IX2-SFR X-Y stage, a U-TVIX-2 camera, and a fluorescence mirror unit cassette for UV/blue/green excitation and detection was employed. The Manders' overlap percentages were executed pixel-by-pixel using the ImageJ software (National Institutes of Health, USA) [81, 82]. Initially, the acquired individual images for the red (anti-HA antibody staining) and green (anti-FRN antibody staining) channels were loaded into ImageJ and converted to RGB format through the menu pathway "Image" > "Type" > "RGB Color". The "Analyze" > "Measure" function was employed to obtain the pixel count for each channel, thereby providing a metric for the total number of pixels present in each channel. To ascertain the colocalisation of the red and green channels, an image calculator tool was utilised via the menu pathway "Process" > "Image Calculator", selecting the "AND" operation to generate a new image depicting the overlapping regions. The pixel count of these overlapping regions was obtained again using the "Analyze" > "Measure" function. The percentage of overlap within the red channel was calculated employing the formula: Percentage of Overlap = (Overlapping Pixels/Total Red Channel Pixels)×100.

## 5.11 Western blot

Before the Western blot analysis, the excision of PolK-loxP parasites was initiated by adding 50 nM rapamycin or DMSO (used as a vehicle control) to synchronized ring-stage cultures with approximately 1% parasitaemia. The cells underwent a 24-hour treatment period, followed by a washout step and cultured again without rapamycin for additional 24h. Then, cultures with 5% parasitaemia at trophozoite/schizont stages were centrifuged in a 15 mL tubes, resuspended in 2 volumes of 0.2% saponin in PBS and incubated on ice for 10 min. Then, 10 mL of PBS was added to each sample and the mixture was centrifuged at 1800 x *g* for 8 min at 4°C. The supernatants were removed and the saponin treatment was repeated two more times. The pellets were transferred to a 1.5 mL vial and washed with PBS, then resuspended in 100 μL of lysis buffer. BioRad Bradford Assay was carried out and 10 μg of each sample was applied in SDS-PAGE gels and then transferred to a PVDF membrane (Bio-Rad, 0.45 μm pore size) by electro-transfer (30 V constant overnight) in the Mini Trans-Blot cell module (Bio-Rad). The membrane was blocked for 1 h at 4°C with 3% (w/v) BSA in PBS-T (PBS plus 0.05% Tween

20) and then incubated for 1 h with a rat anti-HA primary antibody (1:500 [vol/vol] in PBS-T; antibody purchased from Roche, code number 11867423001). After three washing steps PBS-T, a secondary goat anti-rat IgG antibody, HRP conjugated, was used at 1:1000 and incubated for 1 hour. Following three washing steps, the membrane is developed with a chemiluminescent substrate (Super Signal West Pico PLUS) and visualized in ImageQuant LAS 4000 mini Biomolecular Imager (GE Healthcare).

## 5.12 Recombinant expression in yeast

Heterologous expression of *Pf*PolK was performed in *Saccharomyces cerevisiae* W303-1A strain. Cells were routinely cultured in liquid or solid YPD medium (2% dextrose, 2% peptone, 1% yeast extract) or liquid/solid Synthetic Defined medium (SD) without the addition of uracil and with 2% dextrose [83]. Yeasts were transformed with either the empty vector (p416-GPD) or with p416-GPD-PfNF54_070015200 (hereafter referred to as p416-*Pf*PolK, i.e., cloned with the *Plasmodium Pf*PolK gene optimized by Genscript for expression in yeast). Yeast expression vectors were transformed into yeasts by the lithium acetate method [84]. Transformed yeasts were routinely cultured in an SD medium and collected at the early stationary phase for enzymatic assays.

## 5.13 Farnesol and geranylgeraniol kinase activity assays

The recombinant PolK was assayed following the method of Valentin *et al.*, for PhyK assays [35]. For this, yeast crude extracts transformed with p416-*Pf*PolK or the empty vector (control) were employed. Yeasts were cultivated until the stationary phase in SD plus dextrose medium and then cells were disrupted by glass beads (0.5 mm Ø) [85]. Unbroken cells were discarded by centrifugation at 900 x *g* for 1 min and protein was adjusted to 50 mg/ml with 100 mM Tris/HCl pH 7,4. The reaction was performed in 1.5 mL microtubes by incubating approximately 40 mg of yeast protein with 4 mM $MgCl_2$, 800 μM CTP, 10 mM sodium orthovanadate, 0.05% CHAPS and 2 μCi [$^3$H] FOH or [$^3$H] GGOH. [$^3$H] prenol was vacuum-dried as it is commercially distributed in ethanol. The volume was adjusted to 100 μL with 100 mM Tris/HCl pH 7,4 and the reaction was initiated by adding the yeast extract. In some assays, drugs were also added to the reaction or the addition of CTP was omitted as controls. After 30 min of incubation at 37°C, the reaction was stopped by adding 500 μL of n-butanol saturated in water. The mixture was vortexed, and centrifuged at 12.000 x *g* for 10 min and the organic phase was dried under vacuum. The residue was suspended in 10 μL of n-butanol saturated in water and chromatographed on silica 60 plates $F_{254}$ (20x20 cm, Merck). Plates were developed for 7–10 cm with isopropyl alcohol/ammonia (32%)/water (6:3:1 by volume). FOH / GGOH standards and the respective phosphates and pyrophosphates were run on the same plate to identify the reaction products and substrates. Standards were visualized with iodine vapor (See the standards retention in S7 Fig). Finally, the plates were treated with EN3HANCE (Perkin Elmer) and exposed to autoradiography for several days at −70°C. The contrast and brightness of autoradiography scans was adjusted for clarity.

## 5.14 Experiments utilizing human plasma

Specifically, we conducted experiments to verify the fosmidomycin effect in RPMI supplemented with either 10% human plasma or delipidated plasma. In these experiments, we utilized plasma from the blood bank of three different donors and compared fosmidomycin $IC_{50}$ values at 72 h between whole plasma and delipidated plasma from the same donor. Introduction of parasites to RPMI supplemented with plasma or delipidated human plasma occurred at time 0 in the $IC_{50}$ assays. Similar to findings in prior studies by other authors [86]. For all these

experiments, we used AB+ plasma and erythrocytes. The IC$_{50}$ assay was performed as described before and the parasitic growth was monitored with SYBR Green I DNA staining assay [66]. The dilapidation process of human plasma followed the method described by Cham & Knwles further modified by Yao & Vance [87,88]. In this process, 20 mL of plasma was mixed with 10 mg/mL EDTA and 40 mL of a butanol:di-isopropyl either mixture (40:60). The sample was agitated for 30 minutes at 90 rpm and then centrifuged at 2000 x *g* for 2 minutes. The aqueous phase was collected and subjected to an overnight dialysis against 5 L of a 0.9% saline solution using cellulose dialysis tubing (Sigma) to eliminate any remaining traces of solvent. The delipidated plasma was then filter-sterilized and used to prepare complete RPMI medium. Lastly, RPMI containing delipidated human plasma was further enriched with 6 μM cholesterol (Sigma), 30 μM oleic acid (Sigma), and 30 μM palmitic acid (Supelco), all diluted from a x1000-fold concentrated stocks in ethanol [86].

### 5.15 Mass spectrometry analysis

Lipids from 2 mL of plasma or 2 mL of an aqueous solution of 10% AlbuMAX I were extracted following a protocol based on the studies cited herein [40,89,90]. For this, samples were extracted with 18 mL of Chloroform/Methanol (2:1 by volume), mixed, and centrifuged at 1,000 x *g* for 2 minutes. The lower lipid-containing organic phase was dried under nitrogen stream. The residue was dissolved in 1 mL of Ethanol and hydrolyzed with 1 mL of 5M KOH at 56˚C for 1 hour. After cooling and neutralizing with 1 mL of 5M HCl, the solution was partitioned with 4.8 mL of n-Hexane, 1.2 mL of Water, and 1.2 mL of Ethanol. The upper organic phase was evaporated to dryness and resuspended in 100 μL of Ethyl acetate for further analysis. Mass spectrometry analysis was performed following a protocol based on Kai et al. [91]. FOH and GGOH standards at 1 μM were also prepared in Ethyl acetate.

The Gas chromatography–mass spectrometry triple quadrupole (GC-MS/MS) system consisted of a Nexis GC-2030 gas chromatograph coupled with tandem TQ8050 NX mass spectrometer and AOC-5000 auto injector (Shimadzu, Kyoto, Japan). The system was controlled by GC-MS Real Time Analysis software, version 4.53 and the spectra were manipulated using GC-MS Postrun Analysis software, version 4.53. The GC analysis were performed on a Shimadzu column SH-200MS (30m, 0.25 mm I.D., 0.25 μm film thickness), the chromatographic conditions were: inlet temperature 230˚C, pressure 59.8 kPa, total flow 21.6 mL/min, column flow 1.03 mL/min, using helium (purity = 99.999%) as the carrier gas. The temperature was initially held at 60˚C for 1 min, them increased to 160˚C at a rate of 25˚C/min, finally by a 12˚C/min ramp to 300˚C and held by 2 min. Triple-quadrupole MS mass spectrometer was operated at interface and ion source temperatures of 280˚C, solvent delay 3 min and using collision gas argon (minimum purity 99.9999%). The mass spectrometer was working in EI mode of 70 eV. The optimal quantitation and confirmation transitions from parent ions to daughter ions and collision energy for MRM (multiple-reaction monitoring) of each compound were achieved with Auto-MRM study tests performed by the software (MRM transition of GGOH and FOH in the Table 1).

**Table 1. MRM transition of FOH and GGOH..** The table displays the MRM transition of the FOH and GGOH for GC–MS/MS analysis. RT: Retention time, MRM: Multiple reaction monitoring.

| Compound | RT (min) | MRM transition 1 Quantifier | Collision Energy eV | MRM transition 2 Qualifier | Collision Energy eV | MRM transition 3 Qualifier | Collision Energy eV |
|---|---|---|---|---|---|---|---|
| FOH | 7.88 | 93.00>77.00 | 12 | 81.00>79.10 | 9 | 93.00>91.10 | 6 |
| GGOH | 10.97 | 93.00>77.00 | 12 | 81.00>79.10 | 9 | 93.00>91.10 | 6 |

## Supporting information

**S1 Fig.** A) Root mean square deviation (RMSD) values of the protein backbone were used to monitor simulation equilibration and protein changes along the trajectory time (merged 5x1 μs). B) Root mean square fluctuation (RMSF) by residues, calculated using the initial MD frame as a reference and compared between ligand-bound, highlighting the TM2'-TM4 region (the intracellular portion) which displays a unique unfolding in the GGOH simulations. (TIF)

**S2 Fig. Sequence alignment.** ClustalW Multiple alignment of PolK candidate amino acid sequence, predicted in *P. falciparum* NF54 strain against sequences with prenol kinase prediction in UniProt database. PolK: prenol kinase, PhyK: phytol kinase, FolK: farnesol kinase. PfPolK (*P. falciparum* NF54, D0VEH1), SsPhyK (*Synechocystis sp.*, P74653), GmPhyK (*Glycine max*, Q2N2K1), ZmPhyK (*Zea mays*, Q2N2K4), OsPhyK (*Oryza sativa*, Q7XR51), AtPhyK (*Arabidopsis thaliana*, Q9LZ76), AtFolK (*Arabidopsis thaliana*, Q67ZM7). Yellow indicates similarity and red indicates identity. (TIF)

**S3 Fig. Phylogenetic analysis of the retrieved representative prenol binding proteins.** Inset of the overall phylogenetic dendrogram of potential prenol kinases generated using maximum likelihood method (see Methods). Branch support values (Bayes posterior probability) are displayed as numbers for the most relevant clade separation, as well as colours (from the highest scores, in blue, to the lowest values, in red) and thickness of the branches. Organisms and genes from Opisthokont group and some extra outliers are highlighted. (TIF)

**S4 Fig. Phylogenetic analysis of dolichol kinase.** Inset of the overall phylogenetic dendrogram of potential prenol kinases generated using maximum likelihood method (see Methods). Branch support values (Bayes posterior probability) are displayed as numbers for the most relevant clade separation, as well as colours (from the highest scores, in blue, to the lowest values, in red) and thickness of the branches. Apicomplexa clades and the *C. reinhardtii* (A0XX_CHLRE, where A0XX is a generic label for all the *C. reinhardtii*'s taxa) are highlighted. (TIF)

**S5 Fig. Additional Western blot and immunofluorescence experiments.** (A) Photographs of a Western blot of transgenic parasites (left) and the respective PVDF membrane (right) with the protein ladder (M). The Western blot was performed to analyze the HA-tagged PfPolK in parasites where *Pf*PolK was excised (Lane 1, parasites exposed to rapamycin) or preserved (Lane 2, parasites exposed to DMSO). The remaining lanes (group of lanes 3 and onwards) correspond to experiments not related to this article. (B) This panel displays photographs of a Western blot of transgenic parasites (left), the respective Ponceau S staining of the PVDF membrane post-transfer (center), and the protein bands visualized on the Coomassie-stained gel (right). The Precision Plus Protein Kaleidoscop Prestained Protein Standards (BioRad, #1610375) was used to indicate the molecular mass. The Western blot was conducted to analyze the HA-tagged *Pf*PolK in parasites where *Pf*PolK was excised (Lanes 1, 3, and 5, parasites exposed to rapamycin) or preserved (Lanes 2, 4, and 6, parasites exposed to DMSO). (C) Immunofluorescence analysis was conducted on HA-tagged *Pf*PolK parasites. In this analysis, HA-tagged *Pf*PolK is marked in red using α-HA (PolK), the apicoplast is indicated in green using α-FNR (apicoplast), and the nucleus maked in blue using Hoechst 33342 (DNA). The analysis was performed on parasites at the trophozoite stage. (TIF)

**S6 Fig. Recodonized *Pf*PolK sequence.** The sequence displays the cassete composed by the recodonized version of *Pf*Polk (underlined) and a 3x-HA sequence in the 5' end (bold), all flanked by two loxP sites (italics) and the two homology regions upstream and downstream of the cassete (regular). This sequence is part of the transfection plasmid sequence used in this research.
(TIF)

**S7 Fig. Chromatographic analisis of standards.** The images show the retention of different standards in TLC plates, as indicated. Standards were visualized using iodine vapor and UV light.
(TIF)

## Acknowledgments

We thank Prof. Moritz Treeck (The Francis Crick Institute, London, United Kingdom) for providing the Cre-LoxP *P. falciparum* NF54 strain, and Prof. Xavier Fernández-Busquets and Yunuen Avalos Padilla for providing the anti-ferredoxin-NADP reductase antibody and help in IFA experiments. We also thank the Blood Center of Sírio Libanês Hospital (São Paulo, Brazil), for the gift of erythrocytes and plasma, and the CSC-Finland for the generous computational resources.

## Author Contributions

**Conceptualization:** Marcell Crispim, Ignasi Bofill Verdaguer.

**Data curation:** Thales Kronenberger.

**Formal analysis:** Marcell Crispim, Ignasi Bofill Verdaguer, Agustín Hernández, Thales Kronenberger, Àngel Fenollar, Lydia Fumiko Yamaguchi, María Pía Alberione, Miriam Ramirez, Sandra Souza de Oliveira.

**Funding acquisition:** Alejandro Miguel Katzin, Luis Izquierdo.

**Investigation:** Marcell Crispim, Ignasi Bofill Verdaguer, Agustín Hernández, Thales Kronenberger, Àngel Fenollar, Lydia Fumiko Yamaguchi, María Pía Alberione, Miriam Ramirez, Sandra Souza de Oliveira.

**Methodology:** Marcell Crispim, Ignasi Bofill Verdaguer, Agustín Hernández, Thales Kronenberger, Àngel Fenollar, Lydia Fumiko Yamaguchi, María Pía Alberione, Miriam Ramirez, Sandra Souza de Oliveira.

**Project administration:** Alejandro Miguel Katzin, Luis Izquierdo.

**Resources:** Alejandro Miguel Katzin, Luis Izquierdo.

**Software:** Thales Kronenberger.

**Supervision:** Alejandro Miguel Katzin, Luis Izquierdo.

**Validation:** Marcell Crispim, Ignasi Bofill Verdaguer, Agustín Hernández, Thales Kronenberger, Àngel Fenollar, Lydia Fumiko Yamaguchi, María Pía Alberione, Miriam Ramirez, Sandra Souza de Oliveira.

**Visualization:** Marcell Crispim, Ignasi Bofill Verdaguer.

**Writing – original draft:** Marcell Crispim, Ignasi Bofill Verdaguer.

**Writing – review & editing:** Agustín Hernández, Thales Kronenberger, Alejandro Miguel
Katzin, Luis Izquierdo.

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
