## [Decision Letter · Decision Letter 0]

14 Sep 2023

Dear Dr. Katzin,

Thank you very much for submitting your manuscript "Beyond the MEP Pathway: a novel kinase required for prenol utilization by malaria parasites" for consideration at PLOS Pathogens. I apologize for the lengthy time it took to complete the review process. As you are aware, August can be a difficult time to secure reviewers, however the three reviews we obtained were thorough and constructive. 

 As with all papers reviewed by the journal, your manuscript was reviewed by members of the editorial board and by several independent reviewers. In light of the reviews (below this email), we would like to invite the resubmission of a significantly-revised version that takes into account the reviewers' comments.

As you will see from the reviewer's comments below, all three found the identification of this enzymatic activity interesting, in particular because it provides an explanation for previous experimental observations. However, both reviewers 2 and 3 question the overall, biological significance of the findings and explain their logic in detail. This should be explicitly addressed in a revised manuscript if you should choose to submit a revised paper. Reviewer 2 also suggested experiments to confirm conclusions inferred from alpha fold, and both reviewers 2 and 3 requested better data regarding cellular localization of the protein. These concerns will need to be addressed in a revised submission. 

We cannot make any decision about publication until we have seen the revised manuscript and your response to the reviewers' comments. Your revised manuscript is also likely to be sent to reviewers for further evaluation.

Sincerely,

Kirk W. Deitsch

Academic Editor

PLOS Pathogens

James Collins III

Section Editor

PLOS Pathogens

Kasturi Haldar

Editor-in-Chief

PLOS Pathogens

orcid.org/0000-0001-5065-158X

Michael Malim

Editor-in-Chief

PLOS Pathogens

orcid.org/0000-0002-7699-2064

Reviewer's Responses to Questions

**Part I - Summary**

Reviewer #1: The manuscript by Crispim et al. describes the function of a conserved un-annotated Plasmodium gene as being a prenol kinase with the ability to phosphorylate farnesol (FOH) and geranylgeraniol (GGOH). This group has previously reported enzymatic activity in P. falciparum lysate that can produce farnesyl pyrophosphate (FPP) from FOH. This result raises a provocative possibility of a prenol salvage capacity in malaria parasites. Since isopentyl pyrophosphate (IPP), the precursor FPP and other polyprenyl pyrophosphates, is synthesized through methylerythritol phosphate (MEP) pathway, it has been investigated as an attractive target for antimalarial drugs such as fosmidomycin. A salvage bypass, as suggested by the authors, could pose a potential challenge for discovering MEP pathway inhibitors. Thus, the findings are provocative and of considerable significance. There are several minor and moderate concerns that authors need to address. I list them in order of their appearance in the manuscript.

1. Line 27 in Abstract: a fosmidomycin/clindamycin combination is not a “promising” treatment, as evidenced by references cited by the authors on lines 98-101. Authors should moderate this statement and also on lines55 and 56.

2. In Figure 1, DNA gyrase inhibitors should be added along with ribosome inhibitors of apicoplast functions.

3. I did not see that the authors had included Chromera, dinoflagellates and ciliates in their phylogenetic analysis. These are likely to be more closely related to apicomplexans. Also, the statement that apicomplexans are derived from plants and C. reinhardtii (lines 350-353) is incorrect; apicomplexans are members of the Alveolata clade.

4. Figure 2D is repeated twice, the last panel should be “E” (figure 2 and line 218).

5. Line 228: Was an epitope tag added to the gene? It might be worthwhile to state that the gene was codon optimized for yeast expression.

6. Figure 3: What was the level of expression of PfPolK in the yeast? A western blot showing correct size protein expression without degradation should be included.

7. Figure 3 and lines 229-233: Plant phytol phosphate kinase can use any of the nucleotides (ATP, GTP, UTP or CTP) in their phosphorylation reaction. Authors should test the nucleotides other than CTP to see whether it is the only nucleotide used and not others.

8. Lines 249-257: Piggyback insertional mutagenesis screen by Zhang et al. showed this gene to be essential as it resisted being mutagenized. Since this is in contrast to the more robust findings reported here by the authors, it might be worthwhile pointing this out. The failure of Piggyback mutagenesis for this gene may be due to its small size. This addressed in Discussion (Lines 363-365) but might be important to raise in the Results section.

9. Lines 336-340: Sentences are repeated.

10. Lines 350-353; See comment #3 above. These statements are incorrect.

Reviewer #2: In this study, authors identify a novel kinase, PfPolK, and investigate its role in the farnesol (FOH) and geranylgerniol (GGOH) salvage pathway in malaria parasites. The authors use bioinformatic approaches to nominate candidate prenol kinases in Plasmodium falciparum using A. thaliana and Synechocystis spp. as references. Authors then completed a phylogenetic analysis of their candidate kinase, PfPolK, and found that it is most similar to enzymes in unicellular algae. Using Alpha-Fold, authors generate a model to suggesting that the PfPolK has a potential prenol binding pocket that could accommodate both FOH and GGOH. The investigators express PfPolK in S. cerevisiae, and find that yeast lysates from yeast expressing PfPolK can catalyze prenol monophosphorylatation with both FOH and GGOH. Using a conditional knockout of PolK in P. falciparum, the investigators find that the knockout strain has no growth defect, indicating PolK is not an essential enzyme for asexual parasite replication. However, PolK KD parasites no longer are rescued from fosmidomycin treatment by exogenous FOH or GGOH. These findings indicate that PolK may be the kinase responsible for phosphorylating exogenously supplied FOH and GGOH, which can be used to rescue inhibition of isoprenoid metabolism. Finally, authors show that the level of [3H]-FOH incorporation into proteins, as a measure of prenylation, is markedly reduced in the delta-PolK strain, further supporting a role for PolK for incorporation of farnesol.

Overall, this study successfully identified and characterized an enzyme, PolK, that catalyzes phosphorylation of prenols and is required for incorporation of exogenous prenols in Plasmodium falciparum. While this work provides insights into isoprenoid metabolism in P. falciparum, the clinical importance of these findings may be more modest than suggested by the authors. While parasites clearly are capable of prenol alcohol uptake and incorporation into downstream metabolites, it is not clear that they have access to relevant concentrations of prenol alcohols during human infection.

Reviewer #3: This work identifies an enzyme capable of scavenging prenols for the synthesis of isoprenoids in blood-stage malaria parasites. Prenols, particularly genanylgeraniol (GGOH), were known to significantly reduce the sensitivity of P. falciparum parasites to isoprenoid synthesis inhibitors such as fosmidomycin. This phenomenon should require the prenols to be phosphorylated by a prenol kinase followed by formation of prenol pyrophosphates by a second kinase. Crispim and colleagues identified a potential prenol kinase (PolK) and characterized the role of this transmembrane enzyme in blood-stage parasites. Expression in yeast confirmed this enzymatic activity. Deletion of PolK in P. falciparum showed that this enzyme is dispensable, however, the deletion line was more susceptible to fosmidomycin and it could no longer be rescued with prenols. These results demonstrate the enzymatic activity of PolK and show that PolK is required for prenol scavenging, but also show that prenol scavenging is not essential under parasite culture conditions.

**Part II – Major Issues: Key Experiments Required for Acceptance**

Reviewer #1: (No Response)

Reviewer #2: 1) The biological significance of the prenol kinase activity of PolK is not clear. The authors discuss the possibility of dietary acquired FOH/GGOH as a potential mechanism for salvage in malaria parasites as a rationale for studying this protein. However, while the phosphorylated metabolite FPP has been detected in human serum, there is no known dietary derived FOH in human serum, or GGOH. Note that the human Metabolome database has FOH as having been detected in human blood https://hmdb.ca/metabolites/HMDB0004305 However, the sole reference cited (PMID: 9324945) detected FPP in human blood through conversion to FOH. GGOH is not reported to be found in the human metabolome. It seems plausible that the prenol kinase activity of PolK is a moonlighting activity of this enzyme, which may typically have some alternative metabolic function in the parasite. The references provided that suggest a biologically meaningful intake of prenols (e.g., mitigation of statins in a cancer model) do not mitigate this concern, as, for example, PMID: 28710496 employed a murine cancer model in which the diet was supplemented by 0.14 mg/mL GGol, a highly unphysiological concentration. In addition, farnesol is readily oxidized by cytochrome P450 enzymes and glucuronidated, which means that intraerythroycytic P. falciparum is unlikely to encounter this molecule in its unmodified form (PMID: 15320866).

2) Fig. 2 – Conclusions re: AlphaFold modeling are somewhat overstated without experimental validation. For example, the authors propose “a novel prenol binding pocket” (line 359-360). Especially since they have a functional assay in hand, targeted mutagenesis of binding site residues should be used to confirm this model.

3) Figure 4 – quantification and statistical analyses of IF findings shown in panel G should be provided, to support the claim that PolK does not fully co-localize with anti-FNR. A brightfield image would also improve clarity.

4) Discussion Lines 384-402 – this discussion should be softened considerably in the absence of additional evidence for prenol salvage occurring in Plasmodium spp.

5) For all experiments with the PolK knockdown line, methods should clarify duration of culture in the presence/absence of rapamycin.

Reviewer #3: 1. Given the dispensability of PolK and prenol scavenging it is not clear whether PolK is a worthwhile enzyme to target with inhibitors – even in the presence of isoprenoid synthesis inhibitors. Fosmidomycin does work against blood stage parasites in vivo and it is not known whether parasites in red blood cells have sufficient access to serum prenols to significantly affect fosmidomycin activity. One insight into this question could come from using different amounts of human serum (and possibly leukocytes) in the culture system and measure shifts in fosmidomycin susceptibility. Another insight would come from a titration curve showing how much GGOH is required to significantly rescue parasites treated with a concentration of fosmidomycin that can be achieved clinically (10uM?). Without information of this kind, it is hard to assess the possibility that PolK would be of interest to target pharmacologically.

2. A related point connects to the discussion on fosmidomycin sensitivity in the PolK deletion line and the suggestion that prenols in the culture medium (from bovine albumin) may be responsible. They should address the question of bovine albumin being the source by using delipidated albumin reconstituted with oleate and palmitate. As mentioned above, careful attention should be paid to leukocytes. Although the investigators probably remove the buffy coat from the blood they use in culture, they should use a leukodepletion filter to make sure low levels of remaining leukocytes are also removed.

3. The localization of PolK should be further addressed experimentally. The current images are not clear and are not accompanied by quantitative co-localization with apicoplast and ER markers. The discussion of PolK localization at lines 371+2 should also point out that the Boucher used a BioID probe trafficked through the ER, making it potentially challenging to distinguish between ER proteins and apicoplast proteins.

**Part III – Minor Issues: Editorial and Data Presentation Modifications**

Reviewer #1: (No Response)

Reviewer #2: 1. Figure 2 – the overall conclusions of this phylogenetic analysis are relatively modest, especially since many of these enzymes have not been confirmed as prenyl kinases. This figure could be moved to the supplemental data and conclusions softened

2. Figure 3 – it is interesting that this kinase uses CTP. Did the investigators try additional phosphodonor substrates, including other nucleotides? In addition, the position of standards is indicated but should be explicitly shown, ideally on the same TLC plate as the enzyme assays.

3. Figure 4 – Panel F would benefit from reprobing with a control antibody to confirm equal transfer efficiency. The source of anti-HA antibody should be stated. Larger font sizes are suggested to improve readability.

4. Figure 6 – use of prenyl transferase inhibitors to interrupt protein prenylation is recommended as a control to confirm that the incorporation is specific.

5. Even modest reductions in protein prenylation in P. falciparum have been shown to be important for surviving heat shock (PMID: 34182772). Although there is no growth defect under normal asexual replication in the PolK strain, have the authors assessed survival of this strain under heat or cold shock?

Reviewer #3: Fig 3. Standards are not shown for polyprenyl-P and polyprenyl-PP. Densitometry of the bands in the 3 replicate autoradiographs should allow the authors to provide statistics in addition to showing the images for one of the replicates.

Figure 4A. This genotyping strategy does not distinguish between integrants and the transfection plasmid. The insertion should be verified with primer pairs that flank each homology arm to demonstrate integration into the targeted locus.

Figure 4e. The figure legend describes ‘SD of at least three experiments’, but the plots do not seem to have any error bars.

Fig S5. In the Western blot, lower band

Line 177. The word ‘similarity’ in pairwise sequence alignments requires definition. The authors may have meant ‘identity’ here.

Lin 187. Acquisition of PolK through the algal endosymbiont would be most consistent with apicoplast localization of PolK, but this does not seem to be the case (or at least not exclusive apicoplast localization).

Line 198. Replace ‘of the Phe43 and Ile71 to fit the later.’ With ‘of Phe43 and Ile71 to fit the latter.’

Line 238. Autoradiographies -> autoradiographs

Line 252. ‘In contrast’ -> ‘By contrast’

Line 359. Molecular modeling is not conclusive and statements, such as those in line 359 should be moderated (‘the nucleotide ring appears to be stabilized’, ‘The model also suggested a novel’)

Line 401. Change ‘showed to be slightly more susceptible to fosmidomycin.’ To ‘displayed greater sensitivity to fosmidomycin.’

Line 407. Change ‘malaria’ to ‘malaria parasites'

Line 449. The methods state that polyprenyl phosphates were prepared by acid treatment of commercial pyrophosphates. The issue here is that the polyprenyl phosphates are then used as a standard for the enzymatic assays, but there wasn’t clear evidence presented on the synthesis or purity of these polyprenyl phosphate standards.

Line 486. ‘it was calculated the dose-response curve’ -> ‘the dose-response curve was calculated’

Line 495 ‘data was’ -> ‘data were’

The transfection plasmid sequence (particularly the recoded part) should be provided as supplemental material (or a database submission).

PLOS authors have the option to publish the peer review history of their article (what does this mean?). If published, this will include your full peer review and any attached files.

Reviewer #1: No

Reviewer #2: No

Reviewer #3: No
---

## [Decision Letter · Decision Letter 1]

4 Jan 2024

Dear Dr. Katzin,

Thank you very much for submitting your manuscript "Beyond the MEP Pathway: a novel kinase required for prenol utilization by malaria parasites" for consideration at PLOS Pathogens. As with all papers reviewed by the journal, your manuscript was reviewed by members of the editorial board and by several independent reviewers. The reviewers appreciated the attention to an important topic. Based on the reviews, we are likely to accept this manuscript for publication, providing that you modify the manuscript according to the review recommendations.

All three reviewers were quite complementary regarding the extent of the revisions and your attention to their suggestions. They each made additional comments that can be addressed through changes to the text. Reviewer 3 also noted that the diagnostic PCRs regarding the genetic modications should be more accurately performed. This is a relatively simple experiment that will require little effort and should be done. Reviewer 2 suggested experiments involving delipidation of human serum and additional validation of TLC standards. These are valuable suggestions, however given that we do not want to delay publication of the manuscript, addition of these experiments will be left to the authors' discretion. 

Sincerely,

Kirk W. Deitsch

Academic Editor

PLOS Pathogens

James Collins III

Section Editor

PLOS Pathogens

Kasturi Haldar

Editor-in-Chief

PLOS Pathogens

orcid.org/0000-0001-5065-158X

Michael Malim

Editor-in-Chief

PLOS Pathogens

orcid.org/0000-0002-7699-2064

Reviewer Comments (if any, and for reference):

Reviewer's Responses to Questions

**Part I - Summary**

Reviewer #1: Crispim et al. have extensively revised their manuscript in response to its initial review. The manuscript now uses more moderate tone in stating the significance of the findings. In addition, the authors describe new experiments using human plasma that further bolster their suggestion that PfPolK is likely to play phenol salvage role in human infection. Overall, the manuscript is much improved. There may be some minor corrections that may be needed (e.g. Neisseria is NOT an apicomplexan, see line 210.)

Reviewer #2: The authors are applauded in their comprehensive response to my queries and to those of the other reviewers. In particular, the new data is appreciated and helpful. However, a number of queries remain, and I still find many statements to be overly confident or broadly applied, inconsistent with the data presented.

Reviewer #3: The authors did a good job overall of responding to the reviews including providing new data that strengthen the paper. Below are some points to improve the paper with only one substantive point – as described below, the genotyping PCRs are not adequate. The phenotypes described in the paper make it extremely likely that the conditional KO line has the wrong genotype, but it is not a good idea to infer genotypes from phenotypes. It is best practice to fully establish the genotype and use that information to provide explanations for observed phenotypes.

**Part II – Major Issues: Key Experiments Required for Acceptance**

Reviewer #1: (No Response)

Reviewer #2: Experimental concerns:

1. New Figure 7 answers my query as to whether GGOH is found in human serum. However, the authors should confirm that GGOH is depleted by delipidation, since delipidation appears to increase fosmidomycin sensitivity, which they attribute to loss of GGOH.

2. Appreciate the new inclusion of Supplemental Figure S7 to indicate position of standards. However, there can be run to run variability in standards position – it would be better to show these standards on the same TLC plate as used for the enzymatic data shown in Figure 3 (e.g., appreciating that the iodine marks fade, a photo of the same plate including controls such as in the Figure S7). Including the controls as shown confirms the relative positions of the prenol, prenyl-P, and prenyl-PP but does not confirm whether the new bands are in fact the prenyl-P vs. the prenol-PP, since no prenyl-PP is present for comparison. I remain concerns (as per reviewer 3’s query) that the identity and purity of these standards is not certain.

Major concerns with data interpretation/discussion:

3. Discussion paragraph lines 439-443 still overstates the conclusions from the Alpha Fold structural model. Would suggest that the authors make it clear that this is a computational model that awaits experimental validation. In addition, 3. with protein in hand they should be able to determine catalytic efficiency for GGOH vs. FOH substrates directly if this is of interest, so the hypothesizing based on modeled binding energies should be removed.

4. Line 472 – it is not clear from the data provided that this pathway is used in vivo for scavenging. It is still possible that the essentiality of this pathway is limited to the non-physiologic laboratory scenario of isoprenoid inhibition in vitro. The de Wolf paper should not be cited as “dietary sources" of GGol – these animals were forced to ingest high quantities of pure GGol. As written, the authors imply that there are foods that could contain sufficient GGol to complement deficient or inhibited parasites.

5. Discussion paragraph 496-514 still substantially overstates the role of dietary geranylgeraniol. The “cancer therapy” referenced is an HMG CoA reductase inhibitor that specifically inhibits isoprenoid production – the de Wolf study indicates that massive amounts of dietary GGol (0.14 mg/mL body weight) abrogated the antitumor effect in an HMG CoA reductase expressing ovarian cancer model - of course, no data is provided that this in vivo effect is on target. That is hardly generalizable as “implications for cancer therapy” since statins are not used clinically to treat cancer and that amount of dietary GGol is entirely unphysiologic. This is also a preclinical animal model, not a human “clinical trial” as stated in line 501. (The additional 2 references--Prior and Healy--are review articles and do not provide additional experimental evidence to this effect, and should be removed from the reference list in my opinion. As cited, the reader is led to believe that 3 different clinical studies found that dietary GGol impacts clinical cancer treatment, which I find misleading). The authors are additionally directed to an increasing body of literature that suggests a lack of anticancer efficacy of statins, despite the prominence of Ras mutations as oncogenic drivers (e.g., PMID: 31591592).

Reviewer #3: Genotyping PCRs in panel C currently show the absence of the WT gene (with control) and presence of the transgene (with control), but don’t demonstrate that the transgene was inserted in the targeted locus. Loss of the WT amplicon in the transgenic lines is highly suggestive of change at the WT locus, but not sufficient proof (negative data). The PCRs in panel C should have been designed to show insertion into the locus using primers that flank the homology arms. The diCre excision experiments in panel D aren’t the best evidence that the transgene is in the targeted locus, because the primers are in the homology arms (according to the description in the methods). Similar PCR amplicons (including diCre excision) could be obtained from retained transfection plasmid. The conditional KO parasite lines are almost certainly correct, but the genotyping PCRs shown in the paper are not the right way to prove it.

**Part III – Minor Issues: Editorial and Data Presentation Modifications**

Reviewer #1: (No Response)

Reviewer #2: 1. Line 393 – Figure legend for Figure 7 is incorrect. No explanation of A is provided (looks like EIC for GGOH?).

2. Thank you for the image quantification in Figure 4H. I haven’t seen co-localization presented as “percentage overlap” before in other publications. Pearson’s correlation is more commonly used. Can the authors provide justification?

3. A number of minor grammatical mistakes are noted in the discussion paragraph beginning on line 462. Additional copy-editing is recommended.

Reviewer #3: In the response to the reviews, the authors suggest that apicoplast disruption experiments with clindamycin are further evidence that PolK exists outside the apicoplast. I don’t think this point is made in the paper, but it should be modified if it is. Recent studies show that apicoplast enzymes can be active in the vesicles that remain after apicoplast disruption.

In the results and discussion text describing the PiggyBac mutagenesis screen results, be sure use “PiggyBac” rather than “Piggyback”.

Line 209: “Alveolate group,also yielded” Add space before “also”

Line 232: Change “The presumably CTP binding pocket” to use the word “putative” or “predicted” instead of “presumably”.

Figure 5 legend: “was monitored by either flow cytometry.” Is the word “either” needed here? Or, was another technique like SYBR green staining also used?

Line 442: change “presumibly” to “possible” or “predicted”

Line 492: Change “further studies should being carried” to “further studies should be carried”

Line 778: Change “we conducted experiments verify the” to “we conducted experiments to verify the”

PLOS authors have the option to publish the peer review history of their article (what does this mean?). If published, this will include your full peer review and any attached files.

Reviewer #1: No

Reviewer #2: No

Reviewer #3: No

Figure Files:

Data Requirements:

Reproducibility:

References:

---

## [Editor Report · Decision Letter 2]

16 Jan 2024

Dear Dr. Katzin,

We are pleased to inform you that your manuscript 'Beyond the MEP Pathway: a novel kinase required for prenol utilization by malaria parasites' has been provisionally accepted for publication in PLOS Pathogens.

Best regards,

Kirk W. Deitsch

Academic Editor

PLOS Pathogens

James Collins III

Section Editor

PLOS Pathogens

Kasturi Haldar

Editor-in-Chief

PLOS Pathogens

orcid.org/0000-0001-5065-158X

Michael Malim

Editor-in-Chief

PLOS Pathogens

orcid.org/0000-0002-7699-2064
---

## [Editor Report · Acceptance letter]

24 Jan 2024

Dear Dr. Katzin,

We are delighted to inform you that your manuscript, "Beyond the MEP Pathway: a novel kinase required for prenol utilization by malaria parasites," has been formally accepted for publication in PLOS Pathogens.

Best regards,

Michael Malim

Editor-in-Chief

PLOS Pathogens

orcid.org/0000-0002-7699-2064